# LINEAR PROGRAMMING USING DIAGONAL LINEAR NETWORKS

## ABSTRACT

Linear programming has played a crucial role in shaping decision-making, resource allocation, and cost reduction in various domains. In this paper, we investigate the application of overparametrized neural networks and their implicit bias in solving linear programming problems. Specifically, our findings reveal that training diagonal linear networks with gradient descent, while optimizing the squared $L_2$-norm of the slack variable, leads to solutions for entropically regularized linear programming problems. Remarkably, the strength of this regularization depends on the initialization used in the gradient descent process. We analyze the convergence of both discrete-time and continuous-time dynamics and demonstrate that both exhibit a linear rate of convergence, requiring only mild assumptions on the constraint matrix. For the first time, we introduce a comprehensive framework for solving linear programming problems using diagonal neural networks. We underscore the significance of our discoveries by applying them to address challenges in basis pursuit and optimal transport problems.

## 1 INTRODUCTION

Large-scale optimization algorithms play a crucial role in the rapid advancement of modern machine learning and artificial intelligence. Interestingly, these algorithms often introduce an "implicit bias" toward specific solutions, even when such biases are not explicitly defined in the objective or problem formulation. For instance, when tackling the unregularized least squares problem, applying the gradient descent (GD) method typically converges to a solution with minimum Euclidean norm Friedman & Popescu (2004); Yao et al. (2007); Ali et al. (2019), while the coordinate descent method tends to find a solution with the minimum $\ell_1$-norm Gunasekar et al. (2018). These algorithm-induced biases result in a form of "algorithmic regularization" that effectively constrains model complexity. This phenomenon offers insights into the generalization capabilities of deep neural networks trained using (stochastic) gradient descent.

In pursuit of a deeper comprehension of training overparametrized neural networks, recent research has delved into the implicit bias of gradient descent when applied to a reparametrized model Neyshabur et al. (2015); Zhang et al. (2017); Amid & Warmuth (2020a). While traditional gradient descent seeks minimum $\ell_2$-norm solutions in the original space, performing gradient descent within a reparametrized space can induce various other forms of regularization. For instance, in the context of matrix factorization, applying gradient descent with a quadratic reparametrization yields an approximately low-rank solution, provided that the initialization is suitably small Gunasekar et al. (2017); Li et al. (2018). Similarly, when dealing with linear regression, applying gradient descent under a quadratic (or even higher-order) reparametrization with a small initialization typically leads to a sparse solution Vaskevicius et al. (2019); Woodworth et al. (2020); Zhao et al. (2022). Moreover, for a broad range of reparametrizations, it has been established that employing gradient descent with an infinitesimal step size (referred to as gradient flow) is equivalent to utilizing mirror descent (MD) with infinitesimal stepsizes (referred to as mirror flow) and a specific reference divergence Amid & Warmuth (2020b); Azulay et al. (2021); Li et al. (2022).

Most of the existing works on the reparametrized GD focus on the dynamics of GD with infinitesimal stepsizes. This approach provides an elegant characterization of the limit solution given in terms of a regularized optimization problem. In particular, this limit solution is dependent on the size of the initialization. It is not clear how discretization would change the trajectory and the limiting solution

– even basic convergence guarantees seem to be missing in the literature. There are a few recent papers such as Even et al. (2023), in which the authors studied reparametrized GD under a practical stepsize rule. As a tradeoff, these works make some regularity assumptions on the data that may not be readily verified. Therefore, it is a remaining problem to understand the reparametrized GD from a pure algorithmic perspective:

- How does reparametrized GD converge under discretized stepsizes? How to characterize the limiting solution?

In this work, we provide an answer to this question by studying the reparametrized GD under a classical setting of solving a linear program (LP):

$$\min_x \ c^\top x \quad \text{s.t.} \ Ax = b, \ x \geq 0. \tag{1}$$

where $A \in \mathbb{R}^{m \times n}$ with $m \leq n$, $b \in \mathbb{R}^m$, and $c \in \mathbb{R}^n$ satisfies $c > 0$, that is, all elements of $c$ are strictly positive. Note that we assume $c > 0$ for simplicity – see Remark 1.1 that how a general $c$ can be reduced to this case.

OUR CONTRIBUTIONS

• We present a new and simple framework for solving linear programming problems by harnessing the implicit bias of overparametrized neural networks. Our analysis delves into the convergence behaviors of both discrete and continuous-time gradient descent dynamics, establishing a linear rate of convergence. Notably, our investigation of discrete-time dynamics, especially demonstrating global linear convergence (see Theorem 3.4) for diagonal linear networks which we introduce below represents a unique and, to the best of our knowledge, an unprecedented achievement.

• Our work uniquely enables us to elucidate the influence of gradient descent initialization on its convergence to specific solutions. A similar influence on the initialization was shown in Woodworth et al. (2020) for general initialization. However, leveraging this initialization insight, we illustrate how the ultimate outcome of gradient descent dynamics in least squares problems can correspond to the solution of a regularized linear programming problem.

• We conducted a comparative analysis of gradient descent dynamics with mirror gradient descent for the basis pursuit problem and the Sinkhorn algorithm for the optimal transport problem. While there are notable similarities between gradient descent on diagonal networks and these previous algorithms, we demonstrate their distinctions in Section 2.2 and Section 2.3 through explicit elucidation and supported by simulations in Section 4.

Although the formulation (1) was not explicitly discussed in recent literature on reparametrized GD, it includes the well-studied example of sparse linear regression that appeared in Vaskevicius et al. (2019); Woodworth et al. (2020); Zhao et al. (2022), as shown by the following example.

**Example 1.1** *(Basis pursuit) For sparse linear regression with data $X \in \mathbb{R}^{n \times p}$ and $y \in \mathbb{R}^n$, the basis pursuit estimator finds an exact fitting with minimum $\ell_1$-norm:*

$$\min_\beta \ \|\beta\|_1 \quad \text{s.t.} \ X\beta = y. \tag{2}$$

*By a transformation $\beta = w - z$ with both $w \geq 0$ and $z \geq 0$, problem (2) is equivalent to*

$$\min_{w,z} \ 1_n^\top (w + z) \quad \text{s.t.} \ Xw - Xz = y, \ w \geq 0, \ z \geq 0. \tag{3}$$

*which is in the form of (1) with $A = [X, -X]$, $b = y$, $c = 1_{2n}$ and $x = (w; z)$.*

For the sparse linear regression problem, recent literature Vaskevicius et al. (2019); Woodworth et al. (2020); Zhao et al. (2022) considered reparametrizing $\beta = u \circ u - v \circ v$ and running gradient descent steps on the nonconvex loss $\|X(u \circ u - v \circ v) - y\|_2^2$ for variables $u, v \in \mathbb{R}^n$, where $\circ$ denotes the element-wise product of vectors. It was shown that the limiting solution of the GD steps is an approximate solution to (2), given that we take the initialization $u = v = \alpha 1_n$ for some small value $\alpha > 0$ and infinitesimal stepsizes.

In this paper, we adopt a similar reparametrization $x = u \circ u$ for the more general problem (1), and show that the limit of GD on $\|A(u \circ u) - b\|_2^2$ is an approximate solution of (1) for proper initialization. Note that our reparametrization coincides with the one considered in the literature for the basis pursuit problem, following the reduction in Example 1.1. This specific form of reparameterization is commonly referred to as "Diagonal Linear Networks" (DLNs). This quadratic reparameterization is loosely likened to the impact of composing two layers in a neural network. We give a strict analysis of the limiting behavior of GD, without any heuristic assumptions such as the infinitesimal stepsizes or the existence of the limit point – see Sections 2 and 3 for details.

**Example 1.2** *(Optimal transport) Given starting and target distributions $w \in \Delta_m$ and $v \in \Delta_n$ where $\Delta_k := \{x \in \mathbb{R}_+^k \mid 1_k^\top x = 1\}$, and given a cost matrix $C \in \mathbb{R}^{m \times n}$, the optimal transport from $u$ to $v$ is given by the solution of*

$$\min_{X \in \mathbb{R}^{m \times n}} \langle C, X \rangle \quad \text{s.t. } 1_m^\top X = w^\top, \ X1_n = v, \ X \geq 0. \tag{4}$$

*Note that $\langle X, 1_m 1_n^\top \rangle = 1_m^\top X 1_n = 1_m^\top v = 1$. So we can assume $C_{ij} > 0$ for all $(i, j) \in [m] \times [n]$, since otherwise we can add a multiple of $\langle X, 1_m 1_n^\top \rangle$ into the objective.*

Optimal transport is a classical problem in mathematics, economics and operations research dating back to 1780s Monge (1781); Villani et al. (2009). Recently it has regained popularity in the machine learning community with successful applications in computer vision, clustering, sampling etc. For large-scale optimal transport, a popular algorithm is the Sinkhorn algorithm Sinkhorn (1967); Cuturi (2013); Peyré et al. (2019). In particular, the Sinkhorn algorithm finds an approximate solution of (4) with an *entropy regularization*. For the general LP (6), the entropy regularized LP solves the problem

$$\min_z c^\top z + \lambda \sum_{i=1}^n z_i \log(z_i) \quad \text{s.t. } \tilde{A}z = \tilde{b}, \ z \geq 0. \tag{5}$$

where $\lambda > 0$ are fixed parameters. If $\lambda$ is small, the solution of (5) is an approximate solution of (6). See Weed (2018) for a precise analysis. Interestingly, as we show in Sections 2 and 3, using gradient descent on the quadratic reparametrized LP automatically leads to an approximate solution with entropy regularization. See Sections 2.2 and 2.3 for further discussions on the connections of reparametrized GD, mirror descent, and the Sinkhorn algorithm.

In the program (1), the cost vector is assumed to have positive coordinates. This appears to be restricted for general LP with possible negative costs. However, below we show that a general LP can be reduced to the form (1) via a big-M constraint.

**Remark 1.1** *(Reduction of general LP) Consider a general linear program in the standard form*

$$\min_z \tilde{c}^\top z \quad \text{s.t. } \tilde{A}z = \tilde{b}, \ z \geq 0. \tag{6}$$

*where $\tilde{A} \in \mathbb{R}^{m \times n}$, $\tilde{b} \in \mathbb{R}^n$, and where $\tilde{c} \in \mathbb{R}^n$ is a general cost vector, possibly with negative elements. Note that any linear program can be reduced to the above standard form Bertsimas & Tsitsiklis (1997). Suppose (6) has a solution $z^*$ (not necessarily unique), and suppose we can find a number $M$ such that $1_n^\top z^* \leq M$, then problem (6) is equivalent to*

$$\min_{z,t} \tilde{c}^\top z \quad \text{s.t. } \tilde{A}z = \tilde{b}, \ 1_n^\top z + t = M, \ z \geq 0, \ t \geq 0. \tag{7}$$

*We can take $\lambda > 0$ such that $\tilde{c} + \lambda 1_n > 0$. Adding $\lambda$ multiples of the equality constraint $1_n^\top z + t = M$ into the objective, we have an equivalent form*

$$\min_{z,t} (\tilde{c} + \lambda 1_n)^\top z + \lambda t \quad \text{s.t. } \tilde{A}z = \tilde{b}, \ 1_n^\top z + t = M, \ z \geq 0, \ t \geq 0. \tag{8}$$

*The formlation above is in the form of (1) with $A = \begin{bmatrix} \tilde{A} & 0 \\ 1_n^\top & 1 \end{bmatrix}$, $b = [\tilde{b}; M]$, $c = [\tilde{c} + \lambda 1_n; \lambda]$ and $x = [z; t]$.*

By the argument above, a general linear program in standard form can be reduced to the problem (1). In general, the reduction requires finding a big-M constraint that is valid for an optimal solution. For many applications, such a parameter $M$ can be easily computed from $\tilde{A}$ and $\tilde{b}$.

RELATED WORKS

Implicit bias of the training algorithm of a neural network plays a significant role in converging toward a specific global minimum. This has been observed in the past in many occasions in training deep neural network models including Neyshabur et al. (2015); Zhang et al. (2017); Keskar et al. (2017); Soudry et al. (2018) where optimization algorithms, frequently variations of gradient descent, appears to favor solutions with strong generalization properties. This intriguing generalization has also been seen in a rather simpler architecture of neural networks such as diagonal linear networks. The 2-layer diagonal linear network under scrutiny in our study is a simplified neural network that has garnered considerable recent interest Woodworth et al. (2020); Vaskevicius et al. (2019); HaoChen et al. (2021); Pillaud-Vivien et al. (2022). Despite its simplicity, it intriguingly exhibits training behaviors akin to those observed in much more intricate architectures, thus allowing it to be used as a proxy model for a deeper understanding of neural network training. Some prior research Woodworth et al. (2020); Pesme et al. (2021); Nacson et al. (2022) has extensively examined the gradient flow and stochastic gradient flow dynamics in diagonal linear networks, particularly in the context of the basis pursuit problem. These studies have shown that the limit of gradient flow or stochastic gradient flow solves an optimization problem that interpolates between $\ell_1$-norm (the so-called 'rich' regime) and $\ell_2$-norm (neural tangent kernel regime) minimization. While our work shares some commonalities with these findings, we broaden the scope by connecting the limit of continuous and discrete-time gradient descent with a significantly broader class of optimization problems through reevaluation of the influence of initialization on the dynamics. We plan to investigate the impact of step size and small initialization on the generalization properties of the output of diagonal linear networks in future research, building on recent works Even et al. (2023); Pesme & Flammarion (2023); Berthier (2022) in this area.

## 2 REPARAMETRIZED GRADIENT DESCENT

The problem of finding a feasible solution for problem (1) can be formulated as

$$\min_{x \in \mathbb{R}^n} \quad g(x) := \frac{1}{2} \|Ax - b\|_2^2 \quad \text{s.t. } x \geq 0. \tag{9}$$

If problem (1) is feasible, then the optimal value of (9) is 0. Since problem (9) has multiple optimal solutions, using different algorithms for (9) leads to different feasible solutions for (1). In this paper, we consider reparametrizing the nonnegative variable $x = u \circ u$, leading to a non-convex optimization problem:

$$\min_{u \in \mathbb{R}^n} \quad f(u) := \frac{1}{2} \|A(u \circ u) - b\|_2^2 \quad \text{s.t. } u \geq 0. \tag{10}$$

Let $\mathbb{R}_+^n := \{x \in \mathbb{R}^n \mid x \geq 0\}$ and $\mathbb{R}_{++}^n := \{x \in \mathbb{R}^n \mid x > 0\}$. We use gradient descent to solve (10):

---

**Algorithm 1** Gradient Descent for Problem (10)

---

- Initialize from some $u^0 \in \mathbb{R}_{++}^n$.
- For $k = 0, 1, 2, \ldots$ make the updates:

$$u^{k+1} = u^k - \eta_k \nabla f(u^k) = u^k \circ \left( 1_n - 2\eta_k A^\top r^k \right) \tag{11}$$

where $r^k := Ax^k - b$ with $x^k := u^k \circ u^k$, and $\eta_k > 0$ is the stepsize.

---

Note that in the updates (11), the nonnegativity constraint $u \geq 0$ is not explicitly imposed. Therefore, to ensure that the iterates $u^k$ satisfy the nonnegativity constraint, we need to take the stepsizes $\eta_k$ properly. The following lemma provides a practical choice of stepsizes that ensures the nonnegativity of iterates and guarantees the decrease of objective value. Let $L := \|A\|_2^2$ ($\|A\|_2$ is the operator norm of $A$).

**Lemma 2.1** *(Per-iteration decrease) Given any $k \geq 0$, suppose $u^k > 0$, and suppose we take $\eta_k > 0$ such that*

$$\eta_k \leq \min\left\{\frac{1}{4\|A^\top r^k\|_\infty}, \frac{1}{5L\|u^k\|_\infty^2}\right\}. \tag{12}$$

*Then, for all integer $k > 0$, we have $\frac{1}{2}u^k \leq u^{k+1} \leq \frac{3}{2}u^k$, and*

$$f(u^{k+1}) - f(u^k) \leq -\eta_k\|\nabla g(x^k) \circ u^k\|_2^2 = -\frac{\eta_k}{2}\|\nabla f(u^k)\|_2^2. \tag{13}$$

In particular, since $u^0 > 0$, if we take stepsizes $\eta_k$ satisfying the condition (12), then it holds $u^k > 0$ for all $k \geq 0$. Since both the vectors $A^\top r^k$ and $u^k$ are available in the progress of the algorithm, and an upper bound of $L$ can be estimated initially, the stepsize rule (12) is not hard to implement. Suppose the iterations $\{u^k\}_{k\geq0}$ are bounded (which will be formally proved in Section 3), then we can take $\{\eta_k\}_{k\geq0}$ uniformly bounded away from 0 such that (12) is still satisfied.

## 2.1 A CONTINUOUS VIEWPOINT

Before the rigorous analysis of Algorithm 1, we first investigate its continuous version. With infinitesimal stepsizes, the gradient descent updates (11) reduce to the gradient flow:

$$\frac{d}{dt}u(t) = -2u(t) \circ \left(A^\top r(t)\right) \tag{14}$$

where $r(t) = A(u(t) \circ u(t)) - b$. In particular, the integrated form of (14) can be written as

$$u(t) = u(0) \circ \exp\left(-2A^\top \int_0^t r(s)\,ds\right).$$

Therefore, for a positive initialization $u(0) > 0$, the path $u(t)$ will remain in the positive orthant. This is consistent with the result in Lemma 2.1 that the iterates remain positive as long as stepsizes are taken small enough.

The following theorem characterizes the limit of gradient flow under some heuristic assumptions of the convergence.

**Theorem 2.2** *Suppose we initialize the gradient flow (14) with $u(0) = \alpha$ for some $\alpha \in (0, e^{-1})^n$. Suppose $\lim_{t\to\infty} \int_0^t r(s)\,ds$ exists, then $u(t)$ converges as $t \to \infty$. Let $u^* := \lim_{t\to\infty} u(t)$ and $x^* = u^* \circ u^*$, then $x^*$ is the solution of*

$$\min_{x\in\mathbb{R}^n} \quad \sum_{i=1}^n x_i \log\left(\frac{x_i}{\alpha_i^2}\right) - x_i \quad \text{s.t. } Ax = b,\ x \geq 0. \tag{15}$$

*In particular, given $\lambda > 0$, if we take $\alpha_i = \exp(-c_i/(2\lambda))$ for all $i \in [n]$, then $x^*$ is the solution of*

$$\min_{x\in\mathbb{R}^n} \quad c^\top x + \lambda \sum_{i=1}^n \left(x_i \log(x_i) - x_i\right) \quad \text{s.t. } Ax = b,\ x \geq 0. \tag{16}$$

Theorem 2.2 shows an interesting relationship of the limiting solution and the initialization. The limiting solution is an optimal solution of an entropy-regularized LP. The cost of this regularized LP depends on the initialization $\alpha$. In particular, if we take $\alpha$ following $\alpha_i = \exp(-c_i/(2\lambda))$ with a small value of $\lambda$ (which corresponds to small values of $\alpha_i$), then the limiting solution is an approximate solution of the LP (1) since the entropy regularization term in (16) can be made small by choosing $\lambda$ small. On the other hand, if we take larger $\lambda$ (which corresponds to larger initialization), then the entropy term in (16) is not negligible, and the limiting solution is pushed away from the boundary of $\mathbb{R}^n_+$. Such a dependence on initialization has been observed in the special case of basis pursuit Woodworth et al. (2020).

Note that in Theorem 2.2, we have made the technical assumption that $\lim_{t\to\infty} \int_0^t r(s)\,ds$ exists, following a similar assumption in the literature Woodworth et al. (2020). However, as far as we know, this has not been properly justified by a rigorous analysis. In this paper, We give a rigorous analysis directly for the discretized version – see Section 3 for details.

## 2.2 CONNECTIONS TO MIRROR DESCENT

The connections between mirror descent and reparametrized gradient descent have been extensively studied in previous works Amid & Warmuth (2020b); Azulay et al. (2021); Li et al. (2022). When the entropy function is used as a mirror (relative smoothness) function, mirror descent yields multiplicative updates of the iterates . Let $H(x) := \sum_{i=1}^{n} x_i \log(x_i)$ for $x \in \mathbb{R}_+^n$, and let $D_H(x,y) := H(x) - H(y) - \langle \nabla H(y), x - y \rangle$ be the Bregman divergence of $H$. Specifically, for problem (9), the mirror descent has updates:

$$\tilde{x}^{k+1} \in \operatorname*{argmin}_{x \in \mathbb{R}_+^n} \left\{ \langle \nabla g(\tilde{x}^k), x - \tilde{x}^k \rangle + L_k \cdot D_H(x, \tilde{x}^k) \right\} \tag{17}$$

where $L_k > 0$ is a parameter controlling the stepsizes. Equivalently (by the optimality condition),

$$-\nabla g(\tilde{x}^k) = L_k(\nabla H(\tilde{x}^{k+1}) - \nabla H(\tilde{x}^k)) = L_k \log\left(\frac{\tilde{x}^{k+1}}{\tilde{x}^k}\right) = 2L_k \log\left(\frac{\tilde{u}^{k+1}}{\tilde{u}^k}\right) \tag{18}$$

where $\tilde{u}^k := \sqrt{\tilde{x}^k}$. Therefore, the updates in $\tilde{u}^k$ are given by

$$\tilde{u}^{k+1} = \tilde{u}^k \circ \exp\left(-\frac{1}{2L_k} \nabla g(\tilde{x}^k)\right). \tag{19}$$

In particular, if we take $L_k = 1/(2\eta_k)$ with a small value of $\eta_k \to 0$ (i.e. large value of $L_k \to \infty$), then by the first-order Taylor approximation,

$$\tilde{u}^{k+1} \approx \tilde{u}^k \circ \left(1_n - \eta_k \nabla g(\tilde{x}^k)\right). \tag{20}$$

The RHS of (20) is the same as the update in (11). Therefore, with infinitesimal stepsizes, quadratic reparametrized GD for (9) is equivalent to mirror descent with the entropy as the reference function. However, for any finite value of $L_k$, the RHS of (20) is strictly smaller than the RHS of (19), and reparametrized GD is different to mirror descent.

Mirror descent (with entropy as reference function) is also known as exponentiated gradient descent Kivinen & Warmuth (1997); Ghai et al. (2020); Amid & Warmuth (2020a), and both mirror descent and Algorithm 1 belongs to the more general class of multiplicative weight algorithm Arora et al. (2012) – both algorithms involve a rescaling of positive iterates. Although the convergence properties of the mirror descent is well known Beck & Teboulle (2003); Duchi et al. (2010); Lu et al. (2018), the convergence of reparametrized GD is less understood. We provide a rigorous analysis of the convergence of reparametrized GD in Section 3.

## 2.3 CONNECTIONS TO THE SINKHORN ALGORITHM

Algorithm 1 has some similarity to the Sinkhorn algorithm when applied to the optimal transport problem. The Sinkhorn algorithm solves the entropy-regularized LP (5) for the optimal transport problem:

$$\min_X \langle C, X \rangle + \lambda \sum_{i \in [m], j \in [n]} \left(X_{ij} \log(X_{ij}) - X_{ij}\right) \quad \text{s.t. } 1_m^\top X = w^\top, \ X 1_n = v, \ X \geq 0. \tag{21}$$

By the KKT condition of the convex program (21), it can be shown (see e.g. Peyré et al. (2019)) that the (unique) optimal solution $X^*$ has the structure:

$$X_{ij}^* = p_i^* K_{ij} q_j^* \quad \forall i \in [m], \ j \in [n]. \tag{22}$$

where $K_{ij} := e^{-C_{ij}/\lambda}$, and $p^* \in \mathbb{R}_+^m$ and $q^* \in \mathbb{R}_+^n$ are two unknown vectors. The Sinkhorn algorithm initializes with $p^{(0)} = 1_m$, $q^{(0)} = 1_n$, $X^{(0)} = K$, and iteratively rescales the rows and columns to match the constraints:

$$p(t) = \frac{w}{Kq^{(t-1)}}, \quad q^{(t)} = \frac{v}{K^\top p^{(t)}}, \quad X^{(t)} = D(p^{(t)}) K D(q^{(t)}) \tag{23}$$

for $t \geq 1$, where the division is elementwise. It is known that the iterates $p^{(t)}$ and $q^{(t)}$ converge to $p^*$ and $q^*$ respectively, hence $X^{(t)}$ converges to the optimal solution $X^*$. See e.g. Altschuler et al. (2017); Dvurechensky et al. (2018); Lin et al. (2019) for analysis of the converge rate.

It is worth noting that the Sinkhorn algorithm initializes with $X_{ij}^{(0)} := e^{-C_{ij}/\lambda}$, which is the same initialization as Algorithm 1 discussed in Theorem 2.2 in order that the limit solution (for the continuous version) is the entropy regularized LP (16). Moreover, for the optimal transport problem, the updates of Algorithm 1 can be rewritten as

$$(U^{k+1})_{ij} = (U^k)_{ij}\left(1 - \eta_k(g_i + h_j)\right) \tag{24}$$

where $g_i := e_i^\top X^k 1_n - w_i$, $h_j := 1_m^\top X^k e_j - v_j$, and we have used $U^k$ and $X^k$ instead of $u^k$ and $x^k$ to highlight that the variables are matrices. In particular, if the stepsizes $\eta_k$ is very small, then

$$(U^{k+1})_{ij} \approx (U^k)_{ij}\left(1 - \eta_k(g_i + h_j) + \eta_k^2 g_i h_j\right) = (U^k)_{ij}(1 - \eta_k g_i)(1 - \eta_k h_j). \tag{25}$$

Hence the updates above can also be viewed as row and column rescalings, although with different rescaling rules.

Finally, we note that the Sinkhorn updates are equivalent to iterative Bregman projections under KL divergence Benamou et al. (2015), hence in some literature, it is also called a mirror descent type method. See Aubin-Frankowski et al. (2022) for recent results on the connections between the Sinkhorn algorithm and mirror descent.

## 3 CONVERGENCE GUARANTEES

In this section, we provide a rigorous analysis of the convergence of discretized GD for the non-convex problem 10. Problem (10) is a non-convex optimization problem. For a general non-convex optimization problem, it is possible that the GD converges to a saddle point or a local minimum. However, in this section, we prove that GD must converge to the global minimum with a linear rate. Moreover, we will provide a characterization of the limit solution of (discretized) GD, similar to that in Theorem 2.2.

### 3.1 GLOBAL CONVERGENCE

We make the following assumptions on the problem parameters $A$ and $b$.

**Assumption 3.1** *(1) Matrix $A \in \mathbb{R}^{m \times n}$ has full row rank.*

*(2) Problem (1) is strictly feasible, i.e., there exists vector $x \in \mathbb{R}^n$ satisfying $x > 0$ and $Ax = b$.*

Note that Assumption 3.1 (1) is a standard assumption for general linear programs. If $A$ does not satisfy Assumption 3.1 (1), it contains some redundant rows, which can be detected and removed by a basic linear algebra procedure. Assumption 3.1 (2) is also a mild assumption, which is satisfied for most LP problems in practice.

To establish the global convergence of Algorithm 1, we first prove the boundedness of the iterates, given that stepsizes are taken properly.

**Lemma 3.2** *(Boundedness of iterates) Suppose Assumption 3.1 (1) holds true. Suppose we take $\eta_k > 0$ such that (12) holds for all $k \geq 0$. Then it holds*

$$\sup_{k \geq 0} \|u^k\|_2^2 \leq R^2 := \sqrt{n} \max_{\substack{\mathcal{I} \subseteq [n] \\ A_\mathcal{I} \text{ invertible}}} \|(A_\mathcal{I})^{-1}\|_2 \left(\|r^0\|_2 + \|b\|_2\right) + en\|u^0\|_2^2 \tag{26}$$

*where $A_\mathcal{I}$ is the submatrix of $A$ with columns in $\mathcal{I}$.*

Note that the boundedness of the iterates cannot be directly obtained via a level set argument, because the level set of the objective function $f(u) = \|A(u \circ u) - b\|_2^2$ might be unbounded, e.g. in the case when there exists a nonzero vector $x \in \mathbb{R}_+^n$ satisfying $Ax = 0$. Instead, the proof of Lemma 3.2 relies on an in-depth analysis of the dynamic of iterates itself – see Section E for details.

Lemma 3.2 ensures that the iterates will not diverge to infinity, which has an immediate implication on the stepsizes that can be taken.

**Corollary 3.3** *Suppose Assumption 3.1 (1) holds true. Then there exists $\bar{\eta} = \bar{\eta}(A, b, R) > 0$ such that if we take $\eta_k \leq \bar{\eta}$ for all $k \geq 0$, then the conclusion of Lemma 2.1 holds true.*

In particular, we can take constant stepsizes $\eta_k = \eta$ for some $\eta \in (0, \bar{\eta}]$ such that the per-iteration decrease in (13) holds true. In the following, we present the main result of this section under the assumption that constant stepsizes are used.

**Theorem 3.4** *(Global linear convergence) Suppose Assumption 3.1 holds true, and suppose we take $\eta_k = \eta$ for some $\eta \in (0, \bar{\eta}]$ and for all $k \geq 0$. Then there exists a constant $\rho = \rho(A, b, u^0, \eta) \in (0, 1)$ such that*

$$f(u^k) \leq (1 - \rho)^k f(u^0), \quad \forall \, k \geq 1. \tag{27}$$

Theorem 3.4 shows that for the nonconvex problem 10, as long as we take stepsizes small enough, GD always converges to the global optimal solution (which is 0), and has a linear convergence rate. Note that for simplicity, we make the assumption that constant stepsizes are used. The same result still holds true if one uses varying stepsizes $\{\eta_k\}_{k \geq 0}$ satisfying (12) and $\inf_{k \geq 0} \eta_k > 0$. In particular, for the convergence of GD, it is not needed to take diminishing stepsizes that vanish as $k \to \infty$. Note that our analysis is significantly different from (and stronger than) the results that can be obtained by an application of the general convergence results of GD for non-convex problems. See Section A for a discussion.

## 3.2 LIMIT POINT AS AN APPROXIMATE SOLUTION

For the gradient flow, we have shown in Theorem 2.2 that its limiting point is the optimal solution of an entropy-regularized LP. In the following, we show a similar result for the discrete version.

**Theorem 3.5** *Suppose Assumption 3.1 holds true, and suppose we take $u^0 = \alpha \in (0, 1/2)^n$ and $\eta_k = \eta$ for some $\eta \in (0, \bar{\eta}]$ and for all $k \geq 0$. Then the limit $u^\infty := \lim_{k \to 0} u^k$ exists. Let $x^\infty := u^\infty \circ u^\infty$. Then there exists a constant $C = C(A, b, R)$ and a vector $w \in \mathbb{R}^n_+$ with $\|w\|_1 \leq C$ such that $x^\infty$ is the optimal solution of*

$$\min_x \; \sum_{i=1}^n x_i \log\left(\frac{x_i}{\alpha_i^2}\right) - x_i + \eta \log(1/\underline{\alpha}) w_i x_i \qquad \text{s.t. } Ax = b, \; x \geq 0. \tag{28}$$

*where $\underline{\alpha} = \min_{i \in [n]} \alpha_i$. In particular, given $\lambda > 0$, if we take $\alpha_i = \exp(-c_i/(2\lambda))$ and denote $\bar{c} := \max_{i \in [n]} c_i$, then $x^\infty$ is the optimal solution of*

$$\min_x \; c^\top x + \lambda \sum_{i=1}^n \left(x_i \log(x_i) - x_i\right) + \frac{\eta}{2}(\lambda + \bar{c}) w^\top x \qquad \text{s.t. } Ax = b, \; x \geq 0. \tag{29}$$

As shown by Theorem 3.5, if we take the initialization properly, the limit point $x^\infty$ is an optimal solution of the entropy regularized LP given by (29). The objective function in (29) consists of three terms. The first term is the linear objective $c^\top x$ as in (1). The second term is an entropy regularization, which also appears in the limit characterization of the gradient flow (in Theorem 2.2). In particular, if we take a small value of $\lambda$, this entropy regularization is small. The third term is an "error term" from the discrete stepsizes, which does not appear in the counterpart of gradient flow. This error term is proportional to the stepsize $\eta$. If $\eta \to 0$, then this error term vanishes, which is consistent with our result in Theorem 2.2. In the appendix, we provide a more precise characterization of the iteration complexity of obtaining an $\epsilon$-approximation solution using Algorithm 1 – see Lemma J.2.

## 4 EXPERIMENTS

We verify the theoretical findings via simulations. We generate a random matrix $A \in \mathbb{R}^{m \times n}$ with i.i.d. $N(0, 1)$ entries, and a random vector $\bar{x} \in \mathbb{R}^n_+$ with i.i.d. entries following uniform distribution on $[0, 1]$. Then we generate $b := A\bar{x}$. In the following, we set $m = 300$ and $n = 3000$.

**Comparison with mirror descent**. First, we numerically verify the connections between Algorithm 1 and the mirror descent method discussed in Section 2.2. In Section 2.2, it is shown that the

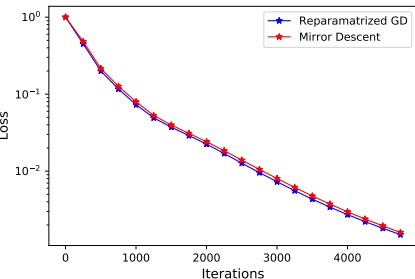 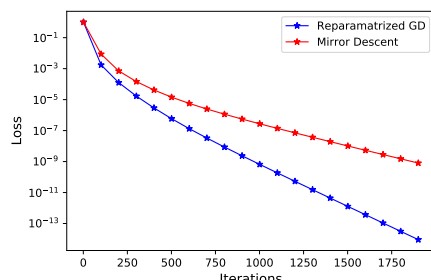

Figure 1: Comparison of reparametrized GD and mirror descent. Left panel: small stepsize. Right panel: large stepsize.

iterates of Algorithm 1 is close to that of mirror descent if the stepsizes are small, while the two algorithms are different for larger stepsizes.

Figure 1 presents iteration-vs-loss plottings of Algorithm 1 and mirror descent under different stepsizes. The y-axis is the normalized loss $\|Ax^k - b\|_2^2/\|b\|_2^2$. Both algorithms are initialized at $x^0 = 10^{-6} \cdot 1_n$. In the left figure, we adopt the stepsizes $\eta_k$ suggested by Lemma 2.1, and set $L_k = 1/(2\eta_k)$ for the mirror descent method. This theoretical choice of $\eta_k$ is very conservative (small). As a result, in the left figure, the iterates of reparametrized GD and mirror descent are very close to each other, which is consistent with the discussions in Section 2.2. On the other hand, the small stepsizes cause slow progress of both algorithms – the loss is quite large even after 5000 iterations. In the right figure, we scale up the stepsizes for both algorithms by a factor of 30. Under this large stepsize, the iterates of reparametrized GD and mirror descent turn out to be significantly different, and both algorithms make fast progress and reach a high accuracy quickly. Reparametrized GD is slightly faster on this example. Moreover, it can be seen that reparametrized GD has an asymptotic linear convergence, which verifies the conclusion of Theorem 3.4.

**Performance under different initializations**. We explore the performance of Algorithm 1 under different initializations. In particular, we consider initializations of the form $u^0 = \alpha 1_n$, where $\alpha > 0$ is a scaler. By Theorems 2.2 and 3.5, under this initialization with a small value of $\alpha$, the limit point of the iterates of Algorithm 1 is an approximate solution of problem (1) for $c = 1_n$. Let $x^*$ be an optimal solution of (1) with $c = 1_n$. Let $\hat{u}$ be the iterate of Algorithm 1 after 5000 iterations, and let $\hat{x} := \hat{u} \circ \hat{u}$. We define the *relative gap* as $1_n^\top(\hat{x} - x^*)/\max\{1, 1_n^\top x^*\}$.

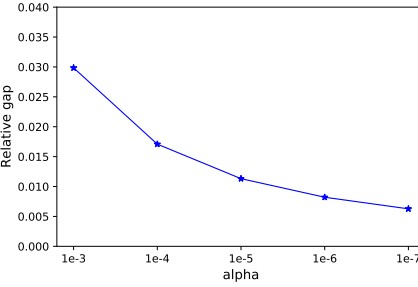 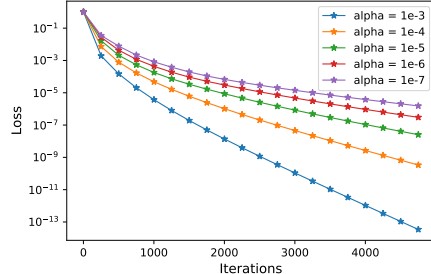

Figure 2: Performance of Algorithm 1 under different sizes of the initialization.

The left panel of Figure 2 presents the relative gap under different values of $\alpha \in \{10^{-3}, 10^{-4}, 10^{-5}, 10^{-6}, 10^{-7}\}$. In particular, for smaller initializations, the limit solution has a smaller relative gap, hence is a better approximate solution of problem (1). But this higher accuracy is not without cost. In the right panel, we present the loss-vs-iteration plottings of Algorithm 1 under these values of $\alpha$. For smaller values of $\alpha$, Algorithm 1 makes slower progress and can only compute a less accurate solution. For instance, with $\alpha = 10^{-3}$, the loss is below $10^{-13}$ after 5000 iterations, but it is only roughly $10^{-5}$ if we set $\alpha = 10^{-7}$.

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

## A  A COMPARISON WITH THE GENERAL CONVERGENCE THEORY OF GD FOR NONCONVEX OBJECTIVES

One intuition for the global convergence of GD stems from the fact that every local minimum of $f(\cdot)$ is a global minimum of $f(\cdot)$ – this can be seen that if $u$ is a local minimum of $f$, then $u \circ u$ is a local minimum of $g$ on $\mathbb{R}^n_+$, and hence a global minimum of $g$ (because $g$ is convex). Moreover, it can be shown that every saddle point of $f(\cdot)$ is a strict saddle point, so intuitively, if the iterates approach any saddle point, it will be pushed away (instead of converging to this saddle). This reminds us of the general convergence theory of GD for smooth nonconvex optimization Lee et al. (2016). In particular, Corollary 9 of Lee et al. (2016) shows that a randomly initialized GD converges to a local minimum almost surely if the objective function only has countable saddle points and every saddle point is strict. Our analysis is significantly different from this result in the following aspects: First, it is possible that the number of saddle points of $f(\cdot)$ is uncountable, hence Corollary 9 of Lee et al. (2016) cannot be directly applied. Second, the results in Lee et al. (2016) hold for a random initialization, while our results hold for an arbitrary initialization $u^0 \in \mathbb{R}^n_{++}$. Finally, Theorem 3.4 proves a linear convergence rate, which cannot be derived from the general results in Lee et al. (2016).

## B  NOTATIONS IN THE PROOFS

Let $\mathbb{R}^n_+ := \{x \in \mathbb{R}^n \mid x \geq 0\}$ and $\mathbb{R}^n_{++} := \{x \in \mathbb{R}^n \mid x > 0\}$. For any $u, v \in \mathbb{R}^n$, let $u \circ v \in \mathbb{R}^n$ be the elementwise multiplication of $u$ and $v$. If all coordinates of $v$ are nonzero, let $\frac{u}{v}$ be the elementwise division of $u$ and $v$. For $u \in \mathbb{R}^n_{++}$ and any univariate function $\varphi$, let $\varphi(u)$ be the vector $[\varphi(u_1), ..., \varphi(u_n)]$. For any matrix $A$, let $\|A\|_2$ be the operator norm of $A$.

For any set $X, Y \subseteq \mathbb{R}^n$, denote $\operatorname{dist}(X, Y) := \inf_{x \in X, y \in Y} \|x - y\|_2$. If $X$ contains only a single point: $X = \{x\}$, then we write $\operatorname{dist}(x, Y) := \operatorname{dist}(\{x\}, Y)$. For any $\epsilon > 0$, define

$$B_\epsilon(X) := \{y \in \mathbb{R}^n \mid \operatorname{dist}(y, X) < \epsilon\}, \quad \bar{B}_\epsilon(X) := \{y \in \mathbb{R}^n \mid \operatorname{dist}(y, X) \leq \epsilon\}. \tag{30}$$

If $X = \{x\}$ (a single point), then we simply write $B_\epsilon(x) := B_\epsilon(X)$ and $\bar{B}_\epsilon(x) := \bar{B}_\epsilon(X)$.

For any vector $x \in \mathbb{R}^n$, let $D(x) \in \mathbb{R}^{n \times n}$ be the diagonal matrix with the diagonal being $x$; let $x_+$ be the vector whose $i$-th element is $\max\{x_i, 0\}$, and let $x_- := x_+ - x$.

## C  PROOF OF LEMMA 2.1

Since $\nabla g$ is $L$-Lipschitz continuous, we have

$$g(x^{k+1}) - g(x^k) \leq \langle \nabla g(x^k), x^{k+1} - x^k \rangle + \frac{L}{2}\|x^{k+1} - x^k\|_2^2. \tag{31}$$

Note that

$$\begin{aligned}
x^{k+1} - x^k &= u^{k+1} \circ u^{k+1} - u^k \circ u^k = (u^{k+1} + u^k) \circ (u^{k+1} - u^k) \\
&= -2\eta_k (u^{k+1} + u^k) \circ u^k \circ \nabla g(x^k) = -2\eta_k D\left((u^{k+1} + u^k) \circ u^k\right) \nabla g(x^k).
\end{aligned} \tag{32}$$

By (31) and (32), and note that $f(u^j) = g(x^j)$, we have

$$\begin{aligned}
f(u^{k+1}) - f(u^k) \leq &-2\eta_k \nabla g(x^k)^\top D\left((u^{k+1} + u^k) \circ u^k\right) \nabla g(x^k) \\
&+ 2L\eta_k^2 \nabla g(x^k)^\top \left[D\left((u^{k+1} + u^k) \circ u^k\right)\right]^2 \nabla g(x^k).
\end{aligned} \tag{33}$$

Note that $\eta_k \leq \frac{1}{4\|A^\top r^k\|_\infty}$ and $u^{k+1} = u^k \circ (1_n - 2\eta_k A^\top r^k)$, so we have

$$\frac{1}{2}u^k \leq u^{k+1} \leq \frac{3}{2}u^k.$$

As a result, we have

$$\begin{aligned}
&2L\eta_k^2 \nabla g(x^k)^\top \left[D\left((u^{k+1} + u^k) \circ u^k\right)\right]^2 \nabla g(x^k) \\
&\leq 2L\eta_k^2 \cdot \frac{5}{2}\|u^k\|_\infty^2 \nabla g(x^k)^\top D\left((u^{k+1} + u^k) \circ u^k\right) \nabla g(x^k) \\
&\leq \eta_k \nabla g(x^k)^\top D\left((u^{k+1} + u^k) \circ u^k\right) \nabla g(x^k),
\end{aligned} \tag{34}$$

where the second inequality made use of $\eta_k \leq \frac{1}{5L\|u^k\|_\infty^2}$. By (33) and (34) we have

$$
\begin{aligned}
f(u^{k+1}) - f(u^k) &\leq -\eta_k \nabla g(x^k)^\top D \left( (u^{k+1} + u^k) \circ u^k \right) \nabla g(x^k) \\
&\leq -\eta_k \nabla g(x^k)^\top D \left( u^k \circ u^k \right) \nabla g(x^k) = -\eta_k \|\nabla g(x^k) \circ u^k\|_2^2.
\end{aligned}
$$

## D   PROOF OF THEOREM 2.2

By the gradient flow (14) we have

$$
u(t) = u(0) \circ \exp\left( -2A^\top \int_0^t r(s)\, ds \right)
$$

where the $\exp(\cdot)$ is applid elementwise. Recall that $x(t) := u(t) \circ u(t)$ and $u(0) = \alpha$, we have

$$
x(t) = (\alpha \circ \alpha) \circ \exp\left( -4A^\top \int_0^t r(s)\, ds \right)
$$

or equivalently,

$$
\log\left( \frac{x(t)}{\alpha \circ \alpha} \right) = -4A^\top \int_0^t r(s)\, ds,
$$

where $\log(\cdot)$ is applid elementwise. Take $t \to \infty$, and let $\nu := -4 \int_0^\infty r(s)\, ds$, we have

$$
\log\left( \frac{x^*}{\alpha \circ \alpha} \right) = A^\top \nu.
$$

Denote $G(x) := \sum_{i=1}^n x_i \log(x_i/\alpha_i^2) - x_i$. Then it can be checked that $\nabla G(x) = \log\left( \frac{x}{\alpha \circ \alpha} \right)$, so we have $\nabla G(x^*) = A^\top \nu$. As a result, $x^*$ satisfies the KKT condition of the problem

$$
\min_{x \in \mathbb{R}^n}\ G(x) = \sum_{i=1}^n x_i \log(x_i/\alpha_i^2) - x_i \quad s.t.\ Ax = b,\ x \geq 0.
$$

## E   PROOF OF LEMMA 3.2

For any vector $v \in \mathbb{R}^m$, let $\mathcal{X}^*(v) := \{x \in \mathbb{R}_+^n \mid Ax = v\}$. Given $k \geq 0$. Let $p^1, ..., p^N$ be the extreme points of $\mathcal{X}^*(Ax^k)$, and let $q^1, ..., q^M$ be the unit vectors of extreme rays of $\mathcal{X}^*(Ax^k)$ (it is possible that $\mathcal{X}^*(Ax^k)$ does not have extreme ray, with $M = 0$). Then we have $p^s \geq 0$ for all $s \in [N]$, and $q^\ell \geq 0$ and $Aq^\ell = 0$ for all $\ell \in [M]$. Moreover, there exist $\{\lambda_s\}_{s=1}^N$ and $\{\mu_\ell\}_{\ell=1}^M$ with $\lambda_s \geq 0$, $\mu_\ell \geq 0$ and $\sum_{s=1}^N \lambda_s = 1$ such that

$$
x^k = \sum_{s=1}^N \lambda_s p^s + \sum_{\ell=1}^M \mu_\ell q^\ell. \tag{35}
$$

Define $p := \sum_{s=1}^N \lambda_s p^s$ and $q := \sum_{\ell=1}^M \mu_\ell q^\ell$, then we have $p \geq 0$, $q \geq 0$, $Aq = 0$, and $x^k = p + q$. By Lemma K.1 we have

$$
4A^\top \left( \sum_{j=0}^{k-1} \eta_j r^j \right) \geq \log\left( \frac{x^k}{x^0} \right) = \log\left( \frac{p + q}{x^0} \right) \geq \log\left( \frac{q}{x^0} \right)
$$

where the last inequality is because $p \geq 0$. Because $q \geq 0$ and $Aq = 0$, the inequality above implies

$$
q^\top \log\left( \frac{q}{x^0} \right) \leq 4q^\top A^\top \left( \sum_{j=0}^{k-1} \eta_j r^j \right) = 0. \tag{36}
$$

Note that for any $i \in [n]$, by Lemma K.6 we know

$$
q_i \log(q_i/(x^0)_i) \geq -(x^0)_i/e. \tag{37}
$$

As a result, for each $t \in [n]$, we have

$$q_t \log(q_t/(x^0)_t) \leq - \sum_{i \in [n] \setminus \{t\}} q_i \log(q_i/(x^0)_i) \leq \sum_{i \in [n] \setminus \{t\}} (x^0)_i/e \leq 1_n^\top x^0/e$$

where the first inequality is by (36), and the second inequality is by (37). Note that if $q_t \geq e(x^0)_t$, then by the inequality above we have $q_t \leq 1_n^\top x^0/e$. So we know

$$q_t \leq \max\left\{ e(x^0)_t, 1_n^\top x^0/e \right\} \leq e 1_n^\top x^0 \quad \forall t \in [n]. \tag{38}$$

On the other hand, for any $s \in [N]$, $p^s$ is an extreme point of $\mathcal{X}^*(Ax^k)$, so there exists $\mathcal{I} \subseteq [n]$ such that $A_\mathcal{I}$ is invertible, and $p^s = (A_\mathcal{I})^{-1}(Ax^k)$. Therefore, we have

$$\|p^s\|_2 \leq \|(A_\mathcal{I})^{-1}\|_2 \|Ax^k\|_2 \leq \max_{\substack{\mathcal{I} \subseteq [n] \\ A_\mathcal{I} \text{ invertible}}} \|(A_\mathcal{I})^{-1}\|_2 \left( \|Ax^k - b\|_2 + \|b\|_2 \right)$$

$$\leq \max_{\substack{\mathcal{I} \subseteq [n] \\ A_\mathcal{I} \text{ invertible}}} \|(A_\mathcal{I})^{-1}\|_2 \left( \|r^0\|_2 + \|b\|_2 \right)$$

where the third inequality made use of $\|Ax^k - b\|_2 \leq \|r^0\|_2$ by Lemma 2.1. As a result,

$$1_n^\top p^s \leq \sqrt{n} \max_{\substack{\mathcal{I} \subseteq [n] \\ A_\mathcal{I} \text{ invertible}}} \|(A_\mathcal{I})^{-1}\|_2 \left( \|r^0\|_2 + \|b\|_2 \right). \tag{39}$$

Making use of (35), (38) and (39), we have

$$1_n^\top x^k = \sum_{s=1}^N \lambda_s (1_n^\top p^s) + 1_n^\top q \leq \sqrt{n} \max_{\substack{\mathcal{I} \subseteq [n] \\ A_\mathcal{I} \text{ invertible}}} \|(A_\mathcal{I})^{-1}\|_2 \left( \|r^0\|_2 + \|b\|_2 \right) + e n 1_n^\top x^0.$$

The proof is complete by noting that $1_n^\top x^k = \|u^k\|_2^2$ and $1_n^\top x^0 = \|u^0\|_2^2$.

## F  Proof of Corollary 3.3

Define

$$\bar\eta := \min_{u \in \bar{B}_R(0)} \left\{ \min \left\{ \frac{1}{4\|A^\top(A(u \circ u) - b)\|_2}, \frac{1}{5L\|u\|_\infty} \right\} \right\}. \tag{40}$$

Then we know $\bar\eta > 0$. By Lemma 3.2 we know $u^k \in \bar{B}_R(0)$ for all $k \geq 0$. Therefore, as long as we take $\eta \in (0, \bar\eta]$ and set $\eta_k = \eta$ for all $k \geq 0$, then the condition (12) holds true, and the conclusion follows Lemma 2.1.

## G  Proof of Theorem 3.4

We first need the following technical result regarding the global convergence of iterates, whose proof is relegated in Section H. Let $\underline{\alpha} := \min_{i \in [n]}(u^0)_i$.

**Lemma G.1** *Suppose Assumption 3.1 holds true, and suppose we take $\eta_k = \eta$ for some $\eta \in (0, \bar\eta]$ and for all $k \geq 0$. For any $\epsilon > 0$, there exists a constant $K = K(A, b, R, \epsilon)$ such that for all $k \geq \lceil (\eta^{-1} + 1)(\log(1/\underline{\alpha}) + 1) \rceil K$, it holds $f(u^k) \leq \epsilon$.*

Our proof of Theorem 3.4 adopts a two-step argument. We first prove a local sub-linear rate (Theorem G.2) using the error bound condition of polynomials. Then using this local convergence rate, we show that the iterates $u^k$ are bounded away from the boundary of $\mathbb{R}_+^n$ (Lemma G.4). Finally, the proof of Theorem 3.4 is complete by a simple application of Lemma G.4.

**Theorem G.2** *(Local convergence: sublinear rate) Suppose Assumptions 3.1 (1) is satisfied, and suppose we take $\eta_k = \eta$ for some $\eta \in (0, \bar\eta]$ and for all $k \geq 0$. Let $\delta = \delta(A, b, R)$ and $\tau = \tau(A, b, R)$ be the constants given by Lemma K.4. Let $k_0 > 0$ be an iteration such that $f(u^{k_0}) \leq \delta$. Then it holds*

$$f(u^k) \leq \left( \delta^{-(1-\tau)} + \frac{\eta c(1-\tau)}{2}(k - k_0) \right)^{-\frac{1}{1-\tau}} \quad \forall k \geq k_0 + 1. \tag{41}$$

*Proof.* By Corollary 3.3 and Lemma K.4 we have

$$f(u^k) - f(u^{k+1}) \geq \frac{\eta}{2}\|\nabla f(u^k)\|_2^2 \geq \frac{\eta c}{2}f(u^k)^{2-\tau}. \tag{42}$$

The conclusion (91) can be derived by using Lemma K.5 with $a_k = f(u^k)$, $K' = k_0$, $\tau' = \tau$ and $c' = \tau c/2$. □

**Corollary G.3** *Under the assumptions of Theorem G.2, it holds*

$$\sum_{k=k_0+1}^{\infty} \|r^k\|_2^2 = 2\sum_{k=k_0+1}^{\infty} f(u^k) \leq \frac{4}{\eta c \tau}\delta^\tau. \tag{43}$$

*Proof.* By (91) we have

$$\begin{aligned}
\sum_{k=k_0+1}^{\infty} f(u^k) &\leq \sum_{k=k_0+1}^{\infty}\left(\delta^{-(1-\tau)} + \frac{\eta c(1-\tau)}{2}(k-k_0)\right)^{-\frac{1}{1-\tau}} \\
&\leq \int_0^\infty \left(\delta^{-(1-\tau)} + \frac{\eta c(1-\tau)}{2}s\right)^{-\frac{1}{1-\tau}}\,ds \\
&= \frac{2}{\eta c(1-\tau)}\int_0^\infty \left(\delta^{-(1-\tau)} + w\right)^{-\frac{1}{1-\tau}}\,dw = \frac{2}{\eta c \tau}\delta^\tau.
\end{aligned}$$

□

**Lemma G.4** *Suppose Assumption 3.1 holds true, and suppose we take $\eta_k = \eta$ for some $\eta \in (0, \bar\eta]$ and for all $k \geq 0$. Then there exists a constant $\sigma = \sigma(A, b, u^0, \eta) > 0$ such that $u^k \geq \sigma 1_n$ for all $k \geq 0$.*

*Proof.* Given any $k \geq 1$, define $v^k := 4\eta\sum_{j=0}^{k-1} r^j$ and $w^k := 16\eta^2\sum_{j=0}^{k-1}(A^\top r^j)\circ(A^\top r^j)$. By Lemma K.1 we have

$$A^\top v^k - w^k \leq 2\log\left(\frac{u^k}{u^0}\right) \leq A^\top v^k.$$

Recall that $x^k = u^k \circ u^k$ and $x^0 = u^0 \circ u^0$, so we have

$$A^\top v^k - w^k \leq \log\left(\frac{x^k}{x^0}\right) \leq A^\top v^k. \tag{44}$$

By Assumption 3.1 (2), there exists a strict feasible point $\tilde x \in \mathbb{R}_{++}^n$ such that $A\tilde x = b$. By (44) we have

$$(\tilde x - x^k)_+^\top \log\left(\frac{x^k}{x^0}\right) \geq (\tilde x - x^k)_+^\top(A^\top v^k - w^k), \tag{45}$$

and

$$-(\tilde x - x^k)_-^\top \log\left(\frac{x^k}{x^0}\right) \geq -(\tilde x - x^k)_-^\top A^\top v^k. \tag{46}$$

Combining (45) and (46) we have

$$\begin{aligned}
(\tilde x - x^k)^\top \log\left(\frac{x^k}{x^0}\right) &\geq (\tilde x - x^k)^\top A^\top v^k - (\tilde x - x^k)_+^\top w^k \\
&= -(r^k)^\top v^k - (\tilde x - x^k)_+^\top w^k
\end{aligned} \tag{47}$$

where the equality made use of $A\tilde x = b$ and $r^k = Ax^k - b$. Note that

$$|(r^k)^\top v^k| = 4\eta\left|(r^k)^\top\sum_{j=0}^{k-1} r^j\right| \leq 4\eta\|r^k\|_2\sum_{j=0}^{k-1}\|r^j\|_2 \leq 4\eta\sum_{j=0}^{k-1}\|r^j\|_2^2 \leq 4\eta\sum_{j=0}^{\infty}\|r^j\|_2^2 \tag{48}$$

where the second inequality made use of the fact that $\|r^j\|_2$ is decreasing (by Lemma 2.1), and

$$\begin{aligned}
\left|(\tilde x - x^k)_+^\top w^k\right| &\leq \|\tilde x - x^k\|_\infty\|w^k\|_1 \leq 16\eta^2\|\tilde x - x^k\|_2\sum_{j=0}^{k-1}\|A^\top r^j\|_2^2 \\
&\leq 16L\eta^2\left(\|\tilde x\|_2 + R\right)\sum_{j=0}^{k-1}\|r^j\|_2^2 \leq 16L\eta^2\left(\|\tilde x\|_2 + R\right)\sum_{j=0}^{\infty}\|r^j\|_2^2.
\end{aligned} \tag{49}$$

Let $C_1 := 4\eta \sum_{j=0}^{\infty} \|r^j\|_2^2 + 16L\eta^2 (\|\tilde{x}\|_2 + R) \sum_{j=0}^{\infty} \|r^j\|_2^2$. Then by Corollary G.3 we know $C_1$ is a finite constant that only depends on $A$, $b$, $u^0$ and $\eta$. As a result of (47), (48), (49) and the definition of $C_1$, we have

$$(\tilde{x} - x^k)^\top \log\left(\frac{x^k}{x^0}\right) \geq -C_1.$$

This implies

$$\tilde{x}^\top \log\left(\frac{x^k}{x^0}\right) \geq (x^k)^\top \log\left(\frac{x^k}{x^0}\right) - C_1 \geq -\frac{n\|x^0\|_1}{e} - C_1 \tag{50}$$

where the second inequality made use of Lemma K.6. Recall that $\underline{\alpha}^2 = \min_{i \in [n]} (x^0)_i$. Note that for any $i \in [n]$,

$$\tilde{x}^\top \log\left(\frac{x^k}{x^0}\right) = \tilde{x}_i \log\left(\frac{(x^k)_i}{(x^0)_i}\right) + \sum_{j \in [n] \setminus \{i\}} \tilde{x}_j \log\left(\frac{(x^k)_j}{(x^0)_j}\right)$$

$$\leq \tilde{x}_i \log\left(\frac{(x^k)_i}{(x^0)_i}\right) + \sum_{j \in [n] \setminus \{i\}} \tilde{x}_j \log\left(\frac{(x^k)_j}{\underline{\alpha}^2}\right) \tag{51}$$

$$\leq \tilde{x}_i \log\left(\frac{(x^k)_i}{(x^0)_i}\right) + \|\tilde{x}\|_\infty \sum_{j \in J} \log\left(\frac{(x^k)_j}{\underline{\alpha}^2}\right)$$

where $J := \{j \in [n] \setminus \{i\} \mid (x^k)_j > \underline{\alpha}^2\}$. We assume $J$ is not empty, otherwise, the second term in the RHS of (51) is 0. By Jensen's inequality, we have

$$\tilde{x}^\top \log\left(\frac{x^k}{x^0}\right) \leq \tilde{x}_i \log\left(\frac{(x^k)_i}{(x^0)_i}\right) + \|\tilde{x}\|_\infty |J| \log\left(\frac{1}{|J|} \sum_{j \in J} \frac{(x^k)_j}{\underline{\alpha}^2}\right)$$

$$\leq \tilde{x}_i \log\left(\frac{(x^k)_i}{(x^0)_i}\right) + n\|\tilde{x}\|_\infty \log\left(\frac{R^2}{\underline{\alpha}^2}\right) \tag{52}$$

where the second inequality is because $n \geq |J| \geq 1$ and $\sum_{j \in J} (x^k)_j \leq \|x^k\|_1 = \|u^k\|_2^2 \leq R^2$ by Lemma 3.2. As a result of (50) and (52), we have

$$\tilde{x}_i \log\left(\frac{(x^k)_i}{(x^0)_i}\right) \geq \tilde{x}^\top \log\left(\frac{x^k}{x^0}\right) - n\|\tilde{x}\|_\infty \log\left(\frac{R^2}{\underline{\alpha}^2}\right)$$

$$\geq -\frac{n\|x^0\|_1}{e} - C_1 - n\|\tilde{x}\|_\infty \log\left(\frac{R^2}{\underline{\alpha}^2}\right). \tag{53}$$

Let $C_2$ be the RHS of (53), then we have

$$(x^k)_i \geq (x^0)_i \exp\left(-\frac{C_2}{\tilde{x}_i}\right) \quad \forall i \in [n].$$

The proof is complete by defining

$$\sigma := \sqrt{\min_{i \in [n]} \left\{(x^0)_i \exp\left(-\frac{C_2}{\tilde{x}_i}\right)\right\}}.$$

$\square$

COMPLETING THE PROOF OF THEOREM 3.4

By Corollary 3.3 we have

$$f(u^k) - f(u^{k+1}) \geq \eta \|u^k \circ \nabla g(x^k)\|_2^2 \geq \eta \sigma^2 \|A^\top (Ax^k - b)\|_2^2$$

$$\geq \mu \eta \sigma^2 \|Ax^k - b\|_2^2 = 2\mu \eta \sigma^2 f(u^k)$$

for any $k \geq 0$, where $\mu$ is the smallest eigenvalue of $AA^\top$, which satisfies $\mu > 0$ because of Assumption 3.1 (1). The proof is complete by defining $\rho := 2\mu \eta \sigma^2$.

# H  PROOF OF LEMMA G.1

Below we first set up some notations and some technical results in Section H.1. Then the proof of Lemma G.1 is presented in Section H.2.

## H.1  NOTATIONS AND TECHNICAL RESULTS

Denote $\mathcal{X}^* := \{x \in \mathbb{R}^n_+ \mid Ax = b\}$ and $\mathcal{U}^* := \{u \in \mathbb{R}^n_+ \mid A(u \circ u) = b\}$. Define the mapping $\varphi(u) := A(u \circ u) - b$. Let $\mathcal{S}$ be the set of stationary points of $f$ in $\mathbb{R}^n_+$, i.e., $\mathcal{S} := \{u \in \mathbb{R}^n_+ \mid \nabla f(u) = 0\}$. For any $\mathcal{I} \in [n]$, let $P_{\mathcal{I}}(b)$ be the projection of $b$ onto the set $\{Ax \mid x \geq 0, x_i = 0 \; \forall i \notin \mathcal{I}\}$. Define

$$\mathcal{R}(b) := \{P_{\mathcal{I}}(b) - b \mid \mathcal{I} \subseteq [n], \; P_{\mathcal{I}}(b) \neq b\}.$$

In particular, note that $\mathcal{R}(b)$ is a finite set.

**Lemma H.1** *For any $\bar{r} \in \mathcal{R}(b)$, there exists $i \in [n]$ such that $(A^\top \bar{r})_i < 0$.*

*Proof.* Suppose (for contradiction) $A^\top \bar{r} \geq 0$ for some $\bar{r} = \mathbb{P}_{\mathcal{I}}(b) - b \in \mathcal{R}(b)$. Let $\bar{x}$ be a point such that $\mathbb{P}_{\mathcal{I}}(b) = A\bar{x}$, then it holds $A^\top(A\bar{x} - b) = A^\top \bar{r} \geq 0$. As a result, $\bar{x}$ is an optimal solution of

$$\min_{x \in \mathbb{R}^n} \; \|Ax - b\|^2_2 \quad \text{s.t.} \; x \geq 0.$$

Therefore, by Assumption 3.1 (2), we know $\bar{r} = A\bar{x} - b = 0$, which is a contradiction to the definition of $\mathcal{R}(b)$. $\qquad\square$

**Lemma H.2** *Let $\mathcal{S}_1 := \mathbb{R}^n_+ \cap \varphi^{-1}(\mathcal{R}(b)) = \{u \in \mathbb{R}^n_+ \mid \varphi(u) \in \mathcal{R}(b)\}$, then it holds $\mathcal{S} \subseteq \mathcal{U}^* \cup \mathcal{S}_1$.*

*Proof.* For any $u \in \mathcal{S}$, it holds

$$\nabla f(u) = 2u \circ [A^\top \varphi(u)] = 0. \tag{54}$$

(Case 1) If $A^\top \varphi(u) = 0$, because $A$ has full row rank (Assumption 3.1 (1)), it holds $u \in \mathcal{U}^*$.

(Case 2) If $A^\top \varphi(u) \neq 0$, let $\mathcal{I} := \{i \in [n] \mid (A^\top \varphi(u))_i = 0\}$. Therefore, by (54), it holds $u_i = 0$ for all $i \in [n] \setminus \mathcal{I}$. By the definition of $\mathcal{I}$ and the KKT condition, we know that $\tilde{x} := u \circ u$ is an optimal solution of

$$\min_x \; \|Ax - b\|^2_2 \quad \text{s.t.} \; x_i = 0 \quad \forall i \in [n] \setminus \mathcal{I}. \tag{55}$$

So we have $P_{\mathcal{I}}(b) = A\tilde{x}$, hence $\varphi(u) = A\tilde{x} - b = P_{\mathcal{I}}(b) - b \neq 0$, where the last inequality is by the assumption of (Case 2). As a result, we have $\varphi(u) \in \mathcal{R}(b)$, and hence $u \in \mathcal{S}_1$. $\qquad\square$

Define

$$\delta_1 = \delta_1(A, b) := \min_{\bar{r} \in \mathcal{R}(b)} \left\{ \max_{i \in [n]} (-A^\top \bar{r})_i \right\} > 0, \tag{56}$$

where $\delta_1 > 0$ is by Lemma H.1 and the fact that $\mathcal{R}(b)$ is a finite set. Let $\delta_2 = \delta_2(A, b) > 0$ be small enough such that

$$\min_{\bar{r} \in \mathcal{R}(b)} \max_{i \in [n]} \inf_{r \in B_{\delta_2}(\bar{r})} (-A^\top r)_i \geq \frac{\delta_1}{2} > 0 \tag{57}$$

and

$$\text{dist}\left(\varphi^{-1}(B_{\delta_2}(\bar{r}^1)), \varphi^{-1}(B_{\delta_2}(\bar{r}^2))\right) \geq \delta_2 \quad \forall \bar{r}^1, \bar{r}^2 \in \mathcal{R}(b). \tag{58}$$

Next, define the set $\Omega_f(\epsilon) := \{u \in \mathbb{R}^n_+ \mid f(u) \geq \epsilon\}$ and

$$\mathcal{F} := \bar{B}_R(0) \bigcap \left[ \bigcap_{\bar{r} \in \mathcal{R}(b)} \left(\varphi^{-1}(B_{\delta_2}(\bar{r}))\right)^c \right] \bigcap \Omega_f(\epsilon) \tag{59}$$

where $R$ is the constant defined in Lemma 3.2. Then we know that $\mathcal{F}$ is a compact set. Moreover, by the definition of $\mathcal{F}$, we know that $\mathcal{S}_1 \cap \mathcal{F} = \emptyset$ and $\mathcal{U}^* \cap \mathcal{F} = \emptyset$. As a result, by Lemma H.2, we know $\mathcal{S} \cap \mathcal{F} = \emptyset$, so $\nabla f(u) \neq 0$ for all $u \in \mathcal{F}$. As a result, we can define

$$\delta_3 = \delta_3(A, b, R, \epsilon) := \min_{u \in \mathcal{F}} \|\nabla f(u)\|_2 > 0. \tag{60}$$

## H.2 Completing the proof of Lemma G.1

First, we have the following claim.

**Claim H.3** *Let $k_0 := \lceil (\eta^{-1} + 1)(\log(1/\underline{\alpha}) + 1) \rceil$ and define*

$$M = M(A, b, R) := \sup_{u \in \bar{B}_R(0)} \|A^\top \varphi(u)\|_2, \quad T = T(A, b, R) := \left\lceil \frac{8}{\delta_1} \max\{1, M\} \right\rceil.$$

*Then for any $k \geq k_0$, at least one of the two cases occurs:*

*(1) $f(u^{Tk}) < \epsilon$.*

*(2) $f(u^k) - f(u^{Tk}) \geq \delta_4$, where*

$$\delta_4 = \delta_4(A, b, R, \epsilon) := \min \left\{ \frac{\delta_3^2}{2}, \frac{1}{2 + 4\bar{\eta}} \delta_2^2, \frac{8\delta_1^2}{81 e^{4M}} \right\}. \tag{61}$$

*Proof of Claim H.3.*

Fix any $k \geq k_0$. Let $\bar{k} := 2(T+1)k$. We assume that (1) does not occur, and prove that (2) occurs. Define $\mathcal{J}_{\mathcal{F}} := \{ j \in [k, \bar{k}) \mid u^j \in \mathcal{F} \}$ and $\mathcal{J}_{\mathcal{F}^c} := \{ j \in [k, \bar{k}) \mid u^j \in \mathcal{F}^c \}$. We discuss a few different cases.

**(Case 1)** $|\mathcal{J}_{\mathcal{F}}| > \eta^{-1}$. By Lemma 2.1, for all $j \in \mathcal{J}_{\mathcal{F}}$, we have

$$f(u^j) - f(u^{j+1}) \geq \frac{\eta}{2} \|\nabla f(u^j)\|_2^2 \geq \frac{\eta}{2} \delta_3^2 \tag{62}$$

where the last inequality is by the definition of $\delta_3$ in (60). As a result,

$$f(u^k) - f(u^{\bar{k}}) \geq \sum_{j \in \mathcal{J}_{\mathcal{F}}} f(u^j) - f(u^{j+1}) \geq \lceil \eta^{-1} \rceil \frac{\eta}{2} \delta_3^2 \geq \frac{\delta_3^2}{2} . \tag{63}$$

**(Case 2)** $|\mathcal{J}_{\mathcal{F}}| \leq \eta^{-1}$. Note that for any $j \in \mathcal{J}_{\mathcal{F}^c}$, it holds $u^j \notin \mathcal{F}$, but we also have $u^j \in \bar{B}_R(0)$ (by Lemma 3.2) and $f(u^j) \geq \epsilon$ (by the assumption that $f(u^{\bar{k}}) \geq \epsilon$). So by the definition of $\mathcal{F}$, we know that there exists $\bar{r} \in \mathcal{R}(b)$ such that $u^j \in \varphi^{-1}(B_{\delta_2}(\bar{r}))$. Below we discuss two cases.

**(Case 2.1)** There exist $\bar{r}^1, \bar{r}^2 \in \mathcal{R}(b)$ with $\bar{r}^1 \neq \bar{r}^2$ such that there exist $j_1, j_2 \in \mathcal{J}_{\mathcal{F}^c}$ with $j_1 < j_2$ and $u^{j_1} \in \varphi^{-1}(B_{\delta_2}(\bar{r}^1))$, $u^{j_2} \in \varphi^{-1}(B_{\delta_2}(\bar{r}^2))$. Since $|\mathcal{J}_{\mathcal{F}}| \leq \eta^{-1}$ (by the assumption of (Case 2)), we can take $j_1 < j_2$ with $j_2 - j_1 \leq \eta^{-1} + 2$. As a result,

$$f(u^{\bar{k}}) - f(u^k) \geq f(u^{j_1}) - f(u^{j_2}) \geq \frac{1}{2\eta} \sum_{j=j_1}^{j_2-1} \|u^{j+1} - u^j\|_2^2 \geq \frac{1}{2\eta(j_2 - j_1)} \left( \sum_{j=j_1}^{j_2-1} \|u^{j+1} - u^j\|_2 \right)^2$$

$$\geq \frac{1}{2\eta(\eta^{-1} + 2)} \|u^{j_1} - u^{j_2}\|_2^2 \geq \frac{1}{2 + 4\bar{\eta}} \delta_2^2$$

where the second inequality is by Lemma 2.1 and the fact $u^{j+1} = u^j - \eta \nabla f(u^j)$; the third inequality is by Jensen's inequality; the fourth inequality is by $j_2 - j_1 \leq \eta^{-1} + 2$ and triangular inequality; the last inequality is by (58).

**(Case 2.2)** There exists $\bar{r} \in \mathcal{R}(b)$ such that $u^j \in \varphi^{-1}(B_{\delta_2}(\bar{r}))$ for all $j \in \mathcal{J}_{\mathcal{F}^c}$. As a result, by (57), there exists $i \in [n]$ such that

$$(A^\top \varphi(u^j))_i \leq -\frac{\delta_1}{2} \quad \forall j \in \mathcal{J}_{\mathcal{F}^c}. \tag{64}$$

Therefore, we have

$$\frac{(u^{j+1})_i}{(u^j)_i} = 1 - \eta(A^\top \varphi(u^j))_i \geq 1 + \frac{\delta_1 \eta}{2} \quad \forall j \in \mathcal{J}_{\mathcal{F}^c}. \tag{65}$$

On the other hand,

$$\frac{(u^{j+1})_i}{(u^j)_i} = 1 - \eta(A^\top \varphi(u^j))_i \geq 1 - \eta M \quad \forall \, j \in \mathcal{J}_\mathcal{F} \text{ or } j \leq k - 1. \tag{66}$$

Combining (65) and (66), and note that $|\mathcal{J}_\mathcal{F}| \leq \lfloor \eta^{-1} \rfloor$, so we have

$$
\begin{aligned}
\frac{(u^{\bar{k}})_i}{(u^0)_i} &= \prod_{j=0}^{\bar{k}-1} \frac{(u^{j+1})_i}{(u^j)_i} \geq (1 - \eta M)^{k + \lfloor \eta^{-1} \rfloor} (1 + \eta \delta_1/2)^{\bar{k} - k - \lfloor \eta^{-1} \rfloor} \\
&\geq (1 - \eta M)^{k + \eta^{-1}} (1 + \eta \delta_1/2)^{\bar{k} - k - \eta^{-1}} \\
&\geq \left[ (1 - \eta M)(1 + \delta_1 \eta/2)^T \right]^{k + \eta^{-1}} \geq \left[ (1 - \eta M)(1 + T \delta_1 \eta/2) \right]^{k + \eta^{-1}},
\end{aligned}
\tag{67}
$$

where the third inequality is because $\bar{k} = 2(T+1)k \geq (T+1)(k + \eta^{-1})$ (since $k \geq k_0 \geq \eta^{-1}$). Recall that $\eta M \leq \bar{\eta} M \leq 1/2$ (by the definition of $\bar{\eta}$ in (40)), we have

$$
\begin{aligned}
(1 - \eta M)(1 + T\delta_1\eta/2) &= 1 - \eta M + \frac{\delta_1 \eta T}{2} - \eta M \cdot \frac{\delta_1 \eta T}{2} \\
&\geq 1 - \eta M + \frac{\delta_1 \eta T}{4} \geq 1 + \frac{\delta_1 \eta T}{8} \geq 1 + \eta,
\end{aligned}
\tag{68}
$$

where the first inequality is because $\eta M \leq \bar{\eta} M \leq 1/2$, then second and third inequalities are by the definition of $T$. As a result of (67) and (68), we have

$$(u^{\bar{k}})_i \geq (u^0)_i(1+\eta)^{k+\eta^{-1}} \geq \underline{\alpha}(1+\eta)^{k_0 + \eta^{-1}} \geq \underline{\alpha}(1+\eta)^{(1+\eta^{-1})\log(1/\underline{\alpha})} \geq \underline{\alpha} e^{\log(1/\underline{\alpha})} = 1, \tag{69}$$

where the second inequality is by the definition of $\underline{\alpha}$ and the assumption $k \geq k_0$; the third inequality is by the definition of $k_0$; the fourth inequality is by the elementary inequality in Lemma K.7.

Now we define

$$\mathcal{J}' := \{ j \in [\bar{k} - 2\lceil \eta^{-1} \rceil - 1, \bar{k} - 1] \mid j \in \mathcal{J}_{\mathcal{F}^c} \}.$$

Recall that $|\mathcal{J}_\mathcal{F}| \leq \lceil \eta^{-1} \rceil$ (by the assumption of (Case 2)), so we know

$$|\mathcal{J}'| \geq \lceil \eta^{-1} \rceil. \tag{70}$$

For any $j \in \mathcal{J}'$, by the assumption of (Case 2.2), we have $u^j \in \varphi^{-1}(B_{\delta_2}(\bar{r}))$, hence by (64) we have $(A^\top \varphi(u^j))_i < -\delta_1/2$. Therefore

$$\|\nabla f(u^j)\|_2 \geq -(\nabla f(u^j))_i = -2(u^j)_i(A^\top \varphi(u^j))_i \geq \delta_1(u^j)_i \quad \forall \, j \in \mathcal{J}'. \tag{71}$$

On the other hand, for any $j \in \mathcal{J}'$,

$$(u^j)_i \geq \frac{u^{\bar{k}}}{(1 + \eta M)^{2\lceil \eta^{-1} \rceil}} \geq \frac{1}{(1 + \eta M)^{2\eta^{-1}+2}} \geq \frac{4}{9e^{2M}}, \tag{72}$$

where the second inequality made use of (69), and the final inequality is because

$$(1 + \eta M)^{2\eta^{-1}} \leq (e^{\eta M})^{2\eta^{-1}} = e^{2M}$$

and $(1 + \eta M)^2 \leq \frac{9}{4}$ (because $\eta M \leq 1/2$). By (71) and (72), we have

$$\|\nabla f(u^j)\|_2 \geq \frac{4\delta_1}{9e^{2M}} \quad \forall \, j \in \mathcal{J}'.$$

As a result,

$$f(u^k) - f(u^{\bar{k}}) \geq \sum_{j \in \mathcal{J}'} f(u^j) - f(u^{j+1}) \geq \sum_{j \in \mathcal{J}'} \frac{1}{2}\eta \|\nabla f(u^j)\|_2^2 \geq \frac{8\delta_1^2}{81e^{4M}}\eta|\mathcal{J}'| \geq \frac{8\delta_1^2}{81e^{4M}},$$

where the last inequality is because of (70).

The proof of Claim H.3 is complete by combining the discussions in (Case 1) and (Case 2).

$\square$

With Claim H.3 at hand we are ready to wrap up the proof of Lemma G.1. Define

$$K = K(A, b, R, \epsilon) := T^\gamma, \quad where \quad \gamma := \lceil f(u^0)/\delta_4 \rceil + 1. \tag{73}$$

Then for any $k \geq Kk_0$, suppose (for contradiction) $f(u^k) > \epsilon$, then $f(u^j) > \epsilon$ for all $j \leq k$, and by Claim H.3 we have

$$f(u^0) - f(u^k) \geq f(k_0) - f(T^\gamma k_0) = \sum_{s=1}^\gamma f(T^{s-1}k_0) - f(T^s k_0) \geq \gamma \delta_4 > f(u^0),$$

which is a contradiction as $f(u^k) \geq 0$. As a result, we know that for any $k \geq Kk_0 = \lceil (\eta^{-1} + 1)(\log(1/\alpha_0) + 1) \rceil K$, it holds $f(u^{\bar{k}}) \leq \epsilon$.

# I  PROOF OF THEOREM 3.5

By the update formula (11) we have

$$u^k = u^0 \circ \prod_{j=0}^{k-1} \left( 1_n - \eta A^\top r^j \right) \tag{74}$$

for all $k \geq 1$. Note that by Theorem 3.4, we know $\sum_{j=0}^\infty \|r^j\|_2 < \infty$, so the update (74) converges as $k \to \infty$.

By Lemma K.1 and recall that $x^k = u^k \circ u^k$ we have

$$4\eta \sum_{j=0}^{k-1} A^\top r^j - 16\eta^2 \sum_{j=0}^{k-1} (A^\top r^j) \circ (A^\top r^j) \leq \log \left( \frac{x^k}{x^0} \right) \leq 4\eta \sum_{j=0}^{k-1} A^\top r^j. \tag{75}$$

Taking $k \to \infty$, and denote $\nu := 4\eta \sum_{j=0}^\infty r^j$ and $\tilde{w} := 16\eta \sum_{j=0}^\infty (A^\top r^j) \circ (A^\top r^j)$, we have

$$A^\top \nu - \eta \tilde{w} \leq \log \left( \frac{x^\infty}{x^0} \right) \leq A^\top \nu. \tag{76}$$

Note that

$$\begin{aligned}
\|\tilde{w}\|_1 &\leq 16\eta \sum_{j=0}^\infty \|A^\top r^j\|_2^2 \leq 16L\eta \sum_{j=0}^\infty \|r^j\|_2^2 \\
&\leq 16L\eta \left( \bar{C} \frac{\log(1/\underline{\alpha})}{\eta} K(A, b, R, \delta) \|r^0\|_2^2 + \frac{2}{\eta c \tau} \delta^\tau \right) \\
&= 16L \log(1/\underline{\alpha}) \left( \bar{C} K(A, b, R, \delta) \|r^0\|_2^2 + \frac{2}{c\tau \log(1/\underline{\alpha})} \delta^\tau \right) \\
&\leq 16L \log(1/\underline{\alpha}) \left( \bar{C} K(A, b, R, \delta) \|r^0\|_2^2 + \frac{2}{c\tau \log(2)} \delta^\tau \right)
\end{aligned} \tag{77}$$

where $K$ is the constant given in Lemma G.1, $\delta$, $c$ and $\tau$ are constants given in Lemma K.4, and $\bar{C}$ is a universal constant. The third inequality is by Lemma G.1 and Corollary G.3; the fourth inequality is by our assumption that $\underline{\alpha} \in (0, 1/2)$. Define

$$C = C(A, b, R) := 16L \left( \bar{C} K(A, b, R, \delta) \|r^0\|_2^2 + \frac{2}{c\tau \log(2)} \delta^\tau \right). \tag{78}$$

Then we have $\|\tilde{w}\|_1 \leq \log(1/\underline{\alpha}) C$. Define

$$w := \left( A^\top \nu - \log \left( \frac{x^\infty}{x^0} \right) \right) \frac{1}{\eta \log(1/\underline{\alpha})}. \tag{79}$$

Then by (76) we have $w \geq 0$, and by (77) and (78) we have $\|w\|_1 \leq \|\tilde{w}\|_1 / \log(1/\underline{\alpha}) \leq C$. Define the function

$$h(x) := \sum_{i=1}^n x_i \log(x_i/\alpha_i^2) - x_i + \eta \log(1/\underline{\alpha}) w_i x_i.$$

Then we have
$$(\nabla h(x^\infty))_i = \log(x_i^\infty/\underline{\alpha}_i^2) + \eta \log(1/\underline{\alpha})w_i = (A^\top \nu)_i,$$
where the second equality is by the definition of $w$ in (79). Moreover, since $Ax^\infty - b = 0$, we know that $x^\infty$ satisfies the KKT condition of the problem (28) (note that the objective function in (28) equals $h$). Therefore it is an optimal solution of (28).

If we take $\alpha_i = \exp(-c_i/(2\lambda))$, then $\log(1/\alpha_i^2) = c_i/\lambda$, and $\log(\underline{\alpha}) = -\frac{\bar{c}}{2\lambda}$. As a result, problem (28) is equivalent to (29).

## J    APPROXIMATION OF LP SOLUTIONS

**Lemma J.1** *Suppose Assumption 3.1 is true. Let*
$$W(\tilde{b}) := \max_x c^\top x \quad s.t. \ Ax = \tilde{b}, \ x \geq 0 \tag{80}$$

*and let $x^*(\tilde{b})$ be the optimal solution. Then there exists $\epsilon_1 > 0$ and $L_1 = L_1(A, b) > 0$ and $L_2 = L_2(A, b) > 0$ such that for any $\tilde{b}$ with $\|\tilde{b} - b\|_2 \leq \epsilon_1$, it holds $|W(b) - W(\tilde{b})| < L_1 \|b - \tilde{b}\|_2$ and $\|x^*(\tilde{b})\|_2 \leq L_2$.*

*Proof.* By duality of linear programs, we have
$$W(\tilde{b}) = \max_y \tilde{b}^\top y \quad s.t. \ A^\top y \leq c \tag{81}$$

Since (by Assumption 3.1) (80) with $\tilde{b} = b$ has a strictly feasible point, we know that there is $\epsilon_1 > 0$ small enough such that for any $\tilde{b}$ with $\|\tilde{b} - b\|_2 \leq \epsilon_1$, the problem (80) is strictly feasible, and hence $W(\tilde{b})$ is a finite value. Let $E$ be the set of extreme points of the polytope $\{y | A^\top \leq c\}$. Then for any $\tilde{b}$ with $\|\tilde{b} - b\|_2 \leq \epsilon_1$, it holds
$$W(\tilde{b}) = \max_y \tilde{b}^\top y \quad s.t. \ y \in E \tag{82}$$

Since $E$ is a finite set, we know that $W$ is a Lipschitz function in the neighborhood of $b$, and the first conclusion holds true. The second conclusion is also true because all coordinates of $c$ is strictly positive. $\square$

**Lemma J.2** *Let $\delta = \delta(A, b, R)$ be the constant given by Lemma K.4, and let $\epsilon_1$ be the constant given by Lemma J.1. Given any $\epsilon \in (0, \min\{\delta, \epsilon_1\})$, suppose we take $\eta = O(\epsilon)$ and $\lambda = O(\epsilon)$, then there is a constant $C = C(A, b)$ and $C' = C'(A, b)$ such that for $k \geq C' \epsilon^{-2}$, we have $\|Ax^k - b\|_2 < \epsilon$, and $|c^\top x^k - c^\top x^*| < C\epsilon$.*

*Proof.* Let $k \geq 1$ be large enough such that $\|Ax^k - b\|_2 < \epsilon$. Denote $\tilde{b}^k := Ax^k$. By a similar argument as in the proof of Theorem 3.5, we know that $x^k$ is the optimal solution of

$$\min_x \ c^\top x + \lambda \sum_{i=1}^n (x_i \log(x_i) - x_i) + \frac{\eta}{2}(\lambda + \bar{c})(w^k)^\top x \tag{83}$$
$$s.t. \ Ax = \tilde{b}^k, \ x \geq 0$$

for some vector $w^k$ satisfying $\sup_{k \geq 1} \|w^k\|_1 \leq C$ for some constant $C = C(A, b, R)$. Let $\tilde{x}^k$ be the optimal solution of
$$\min_x \ c^\top x \quad s.t. \ Ax = \tilde{b}^k, \ x \geq 0 \tag{84}$$

Then by Lemma J.1 we have
$$|c^\top \tilde{x}^k - c^\top x^*| \leq L_1 \|\tilde{b}^k - b^*\|_2 \leq L_1 \epsilon \tag{85}$$

and $\|\tilde{x}^k\|_2 \leq L_2$. Let

$$C_1 := \sup_{x \in \{\tilde{x}^k\} \cup \{x : \|x\|_2 \leq R\}} \left| \sum_{i=1}^n x_i \log(x_i) - x_i \right| \tag{86}$$

and

$$C_2 := \frac{1 + \bar{c}}{2} C(A, b, R) \max\{R, L_2\} \tag{87}$$

Then we have

$$\sup_{x \in \{\tilde{x}^k\} \cup \{x : \|x\|_2 \le R\}} \sup_{\lambda(0,1]} \left| \frac{\lambda + \bar{c}}{2} w^\top x \right| \le C_2 \tag{88}$$

As a result, we have

$$
\begin{aligned}
c^\top \tilde{x}^k \le c^\top x^k \le & c^\top x^k + \lambda \sum_{i=1}^n \left( x_i^k \log(x_i^k) - x_i^k \right) + \frac{\eta}{2} (\lambda + \bar{c})(w^k)^\top x^k + \lambda C_1 + \eta C_2 \\
\le & c^\top \tilde{x}^k + \lambda \sum_{i=1}^n \left( \tilde{x}_i^k \log(\tilde{x}_i^k) - \tilde{x}_i^k \right) + \frac{\eta}{2} (\lambda + \bar{c})(w^k)^\top \tilde{x}^k + \lambda C_1 + \eta C_2 \\
\le & c^\top \tilde{x}^k + 2\lambda C_1 + 2\eta C_2
\end{aligned}
\tag{89}
$$

Suppose we take $\lambda = O(\epsilon)$ and $\eta = O(\epsilon)$, then by (85) and (89) we have

$$|c^\top x^k - c^\top x^*| \le \tilde{C}\epsilon \tag{90}$$

for some constant $\tilde{C} = \tilde{C}(A, b, R)$. Note that as we take $\lambda$ to be small enough, $R$ is uniformly bounded by a value depending on $A$ and $b$, so we can write $\tilde{C} = \tilde{C}(A, b)$.

It remains to find an upper bound on an iteration $k$ such that $\|Ax^k - b\|_2 < \epsilon$. Let $\delta = \delta(A, b, R)$ and $\tau = \tau(A, b, R)$ be the constants given by Lemma K.4. Let $k_0 > 0$ be an iteration such that $f(u^{k_0}) \le \delta$. Then by Theorem G.2 we have

$$f(u^k) \le \left( \delta^{-(1-\tau)} + \frac{\eta c(1 - \tau)}{2} (k - k_0) \right)^{-\frac{1}{1-\tau}} \quad \forall\, k \ge k_0 + 1. \tag{91}$$

Therefore, it takes at most $C'\epsilon^{-2}$ iterations (where $C' = C'(A, b, R)$ is a constant) to reach a point $x^k$ with $f(u^k) = \frac{1}{2}\|Ax^k - b\|_2^2 < \epsilon^2/2$. Again, as we take $\lambda$ to be small enough, $R$ is uniformly bounded by a value depending on $A$ and $b$, so we can write $C' = C'(A, b)$. $\square$

## K   TECHNICAL RESULTS

**Lemma K.1** *Given $K \ge 1$, suppose we take $\eta_k > 0$ (for $k = 0, ..., K - 1$) such that* (12) *holds true for all $k = 0, 1, ..., K - 1$. Then we have*

$$2 \sum_{j=0}^{K-1} \eta_j A^\top r^j - 8 \sum_{j=0}^{K-1} \eta_j^2 (A^\top r^j) \circ (A^\top r^j) \le \log\left( \frac{u^K}{u^0} \right) \le 2 \sum_{j=0}^{K-1} \eta_j A^\top r^j.$$

*Proof.* For any $j \le K - 1$ and $i \in [n]$, By Taylor expansion,

$$\log((u^{j+1})_i) - \log((u^j)_i) = \frac{(u^{j+1})_i - (u^j)_i}{(u^j)_i} - \frac{1}{2} \left( \frac{(u^{j+1})_i - (u^j)_i}{(w^j)_i} \right)^2 \tag{92}$$

where $(w^j)_i$ is between $(u^j)_i$ and $(u^{j+1})_i$. This implies

$$\log((u^{j+1})_i) - \log((u^j)_i) \le \frac{(u^{j+1})_i - (u^j)_i}{(u^j)_i} = 2\eta_j (A^\top r^j)_i. \tag{93}$$

By Lemma 2.1 we know $\frac{1}{2} u^j \le u^{j+1} \le \frac{3}{2} u^j$, so we have $(w^j)_i \ge \frac{1}{2} u^j$, and hence (92) implies

$$
\begin{aligned}
\log((u^{j+1})_i) - \log((u^j)_i) \ge & \frac{(u^{j+1})_i - (u^j)_i}{(u^j)_i} - 2 \left( \frac{(u^{j+1})_i - (u^j)_i}{(u^j)_i} \right)^2 \\
= & 2\eta_j (A^\top r^j)_i - 8\eta_j^2 (A^\top r^j)_i^2.
\end{aligned}
\tag{94}
$$

The proof is complete by summing (93) and (94) over $j$ from 0 to $K - 1$. $\square$

For any $\tilde{b} \in \mathbb{R}^m$, define $\mathcal{X}^*(\tilde{b}) := \{x \in \mathbb{R}_+^n \mid Ax = \tilde{b}\}$.

**Lemma K.2** *(Hoffman constant) There is a constant $H = H(A)$ that only depends on $A$ such that for any $\tilde{b}$ such that $\mathcal{X}^*(\tilde{b}) \neq \emptyset$ and any $x \in \mathbb{R}^n_+$,*

$$\mathrm{dist}(x, \mathcal{X}^*(\tilde{b})) \leq H\|Ax - \tilde{b}\|_2. \tag{95}$$

*Proof.* Define $\bar{A} := [A^\top, -A^\top, -I_n]^\top \in \mathbb{R}^{(2m+n)\times n}$ and $\bar{b} := [\tilde{b}, -\tilde{b}, 0]$, then we can write $\mathcal{X}^*(\tilde{b}) = \{x \in \mathbb{R}^n \mid \bar{A}x \leq \bar{b}\}$. Take $H$ to be the Hoffman constant (see e.g. Pena et al. (2021)) for $\bar{A}$, then we have

$$\mathrm{dist}(x, \mathcal{X}^*(\tilde{b})) \leq H\|(\bar{A}x - \bar{b})_+\|_2 = H\|Ax - \tilde{b}\|_2.$$

$\square$

**Lemma K.3** *(Lojasiewicz's gradient inequality for polynomials) Let $h$ be a polynomial on $\mathbb{R}^n$ with degree $d$. Suppose that $h(0) = 0$ and $\nabla h(0) = 0$, then there exist constants $\bar{c}, \bar{\epsilon} > 0$ such that for all $\|x\|_2 \leq \bar{\epsilon}$ we have*

$$\|\nabla h(x)\|_2 \geq c|h(x)|^{1-\bar{\tau}} \tag{96}$$

*where*

$$\bar{\tau} = \bar{\tau}_{n,d} := \begin{cases} 1, & d = 1, \\ d(3d-3)^{-(n-1)}, & d \geq 2. \end{cases} \tag{97}$$

See Theorem 4.2 of D'Acunto & Kurdyka (2005) for a proof of Lemma K.3.

**Lemma K.4** *(Lojasiewicz's gradient inequality for $f$) There exist constants $\delta = \delta(A, b, R) > 0$, $c = c(A, b, R) > 0$ and $\tau \in (0, 1)$ such that*

$$\|\nabla f(u)\|_2^2 \geq c\Big(f(u)\Big)^{2-\tau} \tag{98}$$

*for all $u \in \mathbb{R}^n_+$ satisfying $f(u) \leq \delta$ and $\|u\|_2 \leq R$.*

*Proof.* Recall that $\mathcal{U}^* = \{u \in \mathbb{R}^n_+ \mid A(u \circ u) = b\}$. Note that $f$ is a polynomial on $\mathbb{R}^n$ with degree 4, and for any $u \in \mathcal{U}^*$, it holds $f(u) = 0$ and $\nabla f(u) = 0$. Therefore, by Lemma K.3, for each $u \in \mathcal{U}^*$, there exists $\bar{\epsilon}_u, \bar{c}_u > 0$ such that

$$\|\nabla f(v)\|_2^2 \geq \bar{c}_u|f(v)|^{2-2\bar{\tau}}, \quad \forall v \in B_{\bar{\epsilon}_u}(u) \tag{99}$$

where $\bar{\tau} = \bar{\tau}_{n,4} = 4 \cdot 9^{-(n-1)}$. Let $\widetilde{\mathcal{U}}^*$ be a compact subset of $\mathcal{U}^*$ such that $\mathrm{dist}(u, \mathcal{U}^*) = \mathrm{dist}(u, \widetilde{\mathcal{U}}^*)$ for all $u$ satisfying $\|u\|_2 \leq R$. Then there is a finite set $\widetilde{\mathcal{U}} \subseteq \widetilde{\mathcal{U}}^*$ such that $\widetilde{\mathcal{U}}^* \subset \cup_{u \in \widetilde{\mathcal{U}}} B_{\bar{\epsilon}_u}(u)$. Moreover, there is $\bar{\epsilon} > 0$ such that $B_{\bar{\epsilon}}(\widetilde{\mathcal{U}}^*) \subset \cup_{u \in \widetilde{\mathcal{U}}} B_{\bar{\epsilon}_u}(u)$. Take $c := \min_{u \in \widetilde{\mathcal{U}}} \bar{c}_u > 0$, we have

$$\|\nabla f(v)\|_2^2 \geq c|f(v)|^{2-2\bar{\tau}}, \quad \forall v \in B_{\bar{\epsilon}}(\widetilde{\mathcal{U}}^*). \tag{100}$$

Let $H$ be the constant given by (K.2), and take $\delta := \bar{\epsilon}^4/(4H^2)$. Then for any $u$ satisfying $\|u\|_2 \leq R$ and $f(u) \leq \delta$, we have

$$\begin{aligned} 2H^2 f(u) =& H^2\|A(u \circ u) - b\|_2^2 \geq \inf_{v \in \mathcal{U}^*} \|u \circ u - v \circ v\|_2^2 \\ =& \inf_{v \in \mathcal{U}^*} \|(u+v) \circ (u-v)\|_2^2 \geq \inf_{v \in \mathcal{U}^*} \|u-v\|_2^4 = \mathrm{dist}(u, \mathcal{U}^*)^4 = \mathrm{dist}(u, \widetilde{\mathcal{U}}^*)^4 \end{aligned} \tag{101}$$

where the first inequality is by Lemma K.2. As a result,

$$\mathrm{dist}(u, \widetilde{\mathcal{U}}^*) \leq (2H^2 f(v))^{1/4} \leq \bar{\epsilon}/2 < \bar{\epsilon}. \tag{102}$$

Therefore, by (100) and (102) for any $u$ satisfying $\|u\|_2 \leq R$ and $f(u) \leq \delta$,

$$\|\nabla f(u)\|_2^2 \geq c|f(u)|^{2-2\bar{\tau}}.$$

The proof is complete by taking $\tau = 2\bar{\tau} = 8 \cdot 9^{-(n-1)}$. $\square$

**Lemma K.5** *Let $\{a_k\}_{k\geq 1}$ be a sequence of decreasing positive numbers, and there are constants $K' > 0$, $c' > 0$ and $\tau' \in (0, 1)$ such that*

$$a_k - a_{k+1} \geq c' a_k^{2-\tau'} \quad \forall k \geq K'. \tag{103}$$

*Then we have*

$$a_k \leq \left[a_{K'}^{-(1-\tau')} + c'(1-\tau')(k - K')\right]^{-\frac{1}{1-\tau'}} \quad \forall k \geq K' + 1.$$

*Proof.* Define $\beta_k = 1/a_k$ for $k \geq K'$. Then (103) becomes

$$\frac{1}{\beta_k} - \frac{1}{\beta_{k+1}} \geq \frac{c'}{\beta_k^{2-\tau'}} \tag{104}$$

Let function $h(\beta) := \beta^{-\frac{1}{1-\tau'}}$ for $\beta \in (0, \infty)$. Note that $h$ is a convex function, so we have

$$\frac{1}{\beta_{k+1}} - \frac{1}{\beta_k} = h(\beta_{k+1}^{1-\tau'}) - h(\beta_k^{1-\tau'}) \geq h'(\beta_k^{1-\tau'})(\beta_{k+1}^{1-\tau'} - \beta_k^{1-\tau'})$$

$$= -\frac{1}{1-\tau'}(\beta_k^{1-\tau'})^{-\frac{1}{1-\tau'}-1}(\beta_{k+1}^{1-\tau'} - \beta_k^{1-\tau'}) = -\frac{1}{1-\tau'}\beta_k^{-(2-\tau')}(\beta_{k+1}^{1-\tau'} - \beta_k^{1-\tau'}) \tag{105}$$

for all $k \geq K'$. By (104) and (105) we have

$$c'\beta_k^{-(2-\tau')} \leq \frac{1}{1-\tau'}\beta_k^{-(2-\tau')}(\beta_{k+1}^{1-\tau'} - \beta_k^{1-\tau'}),$$

which implies

$$\beta_{k+1}^{1-\tau'} - \beta_k^{1-\tau'} \geq c'(1-\tau') \quad \forall k \geq K'.$$

As a result, for any $k \geq K'$, we have

$$\beta_k^{1-\tau'} \geq \beta_{K'}^{1-\tau'} + c'(1-\tau')(k - K')$$

which implies

$$a_k \leq \left[ a_{K'}^{-(1-\tau')} + c'(1-\tau')(k - K') \right]^{-\frac{1}{1-\tau'}} \quad \forall k \geq K' + 1$$

$\square$

**Lemma K.6** *Given $\alpha > 0$, and let $h(t) = t\log(t/\alpha)$ for $t > 0$ and $h(0) = 0$. Then $\min_{t \geq 0} h(t) = -\alpha/e$.*

*Proof.* Let $\bar{t} := \alpha/e$. Then we have $h'(\bar{t}) = \log(\bar{t}/\alpha) + 1 = 0$. Since $h$ is a convex function, so we have

$$h(t) \geq h(\bar{t}) = \frac{\alpha}{e}\log(e^{-1}) = -\frac{\alpha}{e}$$

for all $t \geq 0$. $\square$

**Lemma K.7** *For any $t > 0$, it holds $(1+t)^{1+t^{-1}} \geq e$.*

*Proof.* To prove the conclusion, it suffices to prove that $(1 + t^{-1})\log(1 + t) \geq 1$ for all $t > 0$, or equivalently, proving $(1+t)\log(1+t) - t \geq 0$ for all $t > 0$. Let $h(t) := (1+t)\log(1+t) - t$, then $h'(t) = \log(1+t) > 0$ for all $t > 0$. As a result, $h(t) \geq h(0) = 0$ for all $t > 0$. $\square$

## L  ADDITIONAL EXPERIMENTS

We compare our method against the commercial LP solver Gurobi and another first-order method: an ADMM-based solver SCS. In particular, we simulate the data with the same data-generating process as in Section 4 of the paper, but with $m = 500$ and $n = 50000$. Here is a brief comparison of the methods in this example:

|  | Time (s) | Primal gap | Feasibility gap | Preprocessing time (s) |
|---|---|---|---|---|
| Gurobi | 16.77 | 0 | 0 | 10.11 |
| SCS 100 iterations | 40.49 | -0.971 | 2.21e-10 | - |
| SCS 200 iterations | 65.06 | 0.008 | 1.57e-12 | - |
| Alg1 1000 iterations | 9.26 | 0.005 | 4.17e-11 | - |

where the primal gap is defined as $(c^\top \hat{x} - c^\top x^*)/c^\top x^*$, with $\hat{x}$ be the solution by an algorithm and $x^*$ is the optimal solution (by Gurobi). The feasibility gap is defined as $\|A\hat{x} - b\|_2^2 / \max\{1, \|b\|_2^2\}$.

First, it can be seen that our method is much faster than SCS, where our method computes a solution with a smaller primal gap in a much shorter time. Our method is also faster than Gurobi if a low-accuracy solution with a primal gap 0.005 is satisfactory for the underlying application. Moreover, we note that Gurobi takes 10.11 seconds for preprocessing, which is already similar to the runtime of our method.

Admittedly, our method is not able to obtain high-accuracy solutions as fast as Gurobi, but it is useful for quickly obtaining an approximate solution. Moreover, since Algorithm 1 only involves matrix-vector multiplications, it is easily implemented in a GPU-acceleration setting, which can potentially outperform Gurobi for larger problems.

