# OpenReview forum: "Linear programming using diagonal linear networks"
_ICLR.cc/2024/Conference — Submitted to ICLR 2024_

### Official Review · Reviewer_9DAk · 2023-10-29

**Soundness:** 2 fair
**Presentation:** 3 good
**Contribution:** 2 fair
**Rating:** 6
**Confidence:** 4

**Summary:**

This paper shows that gradient descent on a diagonal linear network (with a quadratic parametrization) under specific initialization leads to an approximate solution to the entropy-regularized solution to a linear programming problem. The convergence results are presented for both gradient flow and gradient descent algorithms.

**Strengths:**

1. This paper presents a new idea that solves LPs via training a diagonal linear network, which exploits the implicit bias of diagonal networks.
2. linear convergence results are shown for training diagonal linear networks with GD, which is different from what has been established, for example, the results in Vaškevičius et. al. 2019.

**Weaknesses:**

The main weaknesses are the presentation and the significance of the results, mainly for Theorem 3.4 and 3.5.
1. There needs more discussion on the upper bound $\bar{\eta}$ on the step size, and the linear rate $\rho$ in Theorem 3.4, specifically their dependence on 1) the underlying LP problem $(A,b,c)$, and its scale (# of decision variables, #of constraints, etc.); 2) the initialization $u_0$. For example, if either $A$ is ill-conditioned, or the initialization is close to the origin, then I believe $\rho$ should be close to one. Merely showing that GD converges linearly does not make a significant contribution if what authors propose is to implement this GD algorithm for solving real LP problems.
2. Theorem 3.5 only shows that the GD converges to some $x^\infty$ that is close to the desired solution to the LP, but the result is weak in the sense that it doesn't suggest an upper bound on the # of GD iterations for achieving certain accuracy. Specifically, the convergence result one expects is that given some $\epsilon>0$, the GD with some step size $\eta(\epsilon)$ takes $T(\epsilon)$ iterations to achieve either 1) $\|x^T-x^*\|\leq \epsilon$, where $x^*$ is the true optimal solution; 2) or the optimality gap is less than $\epsilon$.
3. Another concern I have is that I don't find, from the discussions and experiments in this paper, any evidence that the proposed algorithm has advantages in solving certain LPs, compared to existing methods.

**Questions:**

See "weaknesses"

---

> ### Author Response · Authors · 2023-11-18
> **Rebuttal (Thanks for the review and incisive questions. We truly appreciate your comments.)**
>
> $\textbf{Response on the comments on stepsize and linear rate:}$ The upper bound on $\eta$ was explicitly written down in equation (6.1) in the appendix
> 		$$\eta := \min_{ u\in  B_R(0) }  \Big(   \min \big(  \frac{1}{4\text{Res}}, \frac{1}{5L|u|_{\infty}}  \big)  \Big) $$ where $\text{Res}:= |A^\top( A(u\circ u)-b )|_2$ and the value of $R$ can be large in the worst case (leading to a conservative constant for the convergence rate), it can be much better in cases where the matrix $A$ satisfies some regularity properties. For example, if all the entries of $A$ are i.i.d. Gaussian, then one can show that with high probability, $R$ has a polynomial dependence on the problem sizes.
>
> The dependence of $\rho$ on $A$, $b$ and the initial point can be derived from the proof of Theorem 3.4. This dependence is hard to be written in a clean way, as we have used few constants that can be shown to exist but cannot be written in a closed form (see Appendix H.1). Furthermore, since our problem is non-convex in nature, it makes it lot more harder to delineate exact expression of the linear rate. However, we agree that if $A$ is ill-conditioned or the initial point is close to $0$, the value of $\rho$ is close to $1$.
>
> Finally, we note that conservative constants (e.g. Hoffman constant) have been commonly used in deriving the linear convergence of first order methods for solving LP problems.  Although such constants may not reflect the actual performance of the algorithm, it provides some hint on the convergence properties of the algorithm. Some of the impactful papers from the past with similar problem includes [1] and [2].
>
> $\textbf{Response on the dependence of stepsize and running time on optimality gap:}$ This is an excellent question. We definitely like to include more details in the Appendix. To emphasize how the stepsize depends on the gap, let us define
>   $$  C_1 := \sup_{ x \in (x^*) \cup (x: \|x\|_2 \leq R) } \Big| \sum^n x_i \log(x_i) - x_i \Big|, \quad   C_2 := \frac{1+ c}{2} \max ( R, \| x^*\|_2 )
>   $$
> where $x^*$ is the solution of the LP problem and $R$ is the same constant as in the response of the previous comment. Now a simple but tedious computation would show that if the regularization $\lambda(\epsilon)$ and stepsize $\eta(\epsilon)$ is chosen in way so that $2\lambda(\epsilon) C_1 + 2\eta(\epsilon) C_2 <\epsilon$, then the optimality gap after large enough time $T$ will be less than $\epsilon$.
>
> The a number of iterations $T(\epsilon)$ to reach an $\epsilon$-optimality gap is more subtle.
> Since our analysis here is significantly different than usual first order methods, it is hard to write a closed-form expression of $T(\epsilon)$. This is partly because our results shows global convergence rate of GD for a non-convex problem. Unlike the analysis of a convex problem, the typical per-iteration analysis is not working here. We have to define some global constants that can be shown to exist but cannot be written in a closed form (see Appendix H.1). The dependence on $\epsilon$ is therefore encoded in a sophisticated way.
>
> $\textbf{Response on comparison with other commercial LP solver:}$  We have run a new experiment comparing our method against
>  the commercial LP solver Gurobi and another first-order method: an ADMM-based solver SCS. In particular, we simulate the data with the same data-generating process as in Section 4 of the paper, but with $m = 500$ and $n = 50000$. Here is a brief comparison of the methods in this example: Gurobi, SCS and our algorithm achieve respectively 0, -0.971,0.005 primal gaps and 0, 2.21e-10, 4.17e-11 feasibility gaps in 16.77, 40.49 and 9.26 seconds.
>
> Here the primal gap is defined as $(c^\top \hat x - c^\top x^*) / c^\top x^*$, with $\hat x$ be the solution by an algorithm and $x^*$ is the optimal solution (by Gurobi). The feasibility gap is defined as $\| A \hat x - b \|_2^2 / \max\{1, \| b\|_2^2 \} $.
> First, it can be seen that our method is much faster than SCS,
> where our method computes a solution with a smaller primal gap in a much shorter time.
> Our method is also faster than Gurobi if a low-accuracy solution with a primal gap $0.005$ is satisfactory for the underlying application.
> Moreover, we note that Gurobi takes $10.11$ seconds for preprocessing, which is already similar to the runtime of our method.
>
> Admittedly, our method is not able to obtain high-accuracy solutions as fast as Gurobi, but it is useful for quickly obtaining an approximate solution. Moreover, since Algorithm 1 only involves matrix-vector multiplications, it is easily implemented in a GPU-acceleration setting, which can potentially outperform Gurobi for larger problems.
>
> [1] Hong, M., Luo, Z.-Q. On the,  linear convergence of the alternating direction method of multipliers. (2019) Mathematical Programming.
>
> [2] Luo, Z.-Q., Tseng, P. On the linear convergence of descent methods for convex essentially smooth minimization (1992) SIAM Journal on Control and Optimization

---

> > ### Author Response · Authors · 2023-11-20
> > **update/feedback**
> >
> > Dear Reviewer 9DAk,
> >
> > As the deadline for the discussion phase is fast approaching, we were curious if you had any feedback for our response. Please also let us know if you have any further questions. Thank you and looking forward to hearing from you.
> >
> > Sincerely,
> >
> > Authors.

---

> > > ### Comment · Reviewer_9DAk · 2023-11-20
> > > **Thanks for the response**
> > >
> > > The discussion on $T(\epsilon)$ is helpful but not fully satisfactory. My main concern is that the paper should ideally show the computation complexity for getting an $\epsilon$-optimal solution to their regularized LP problem, but neither Theorem 3.4 nor Theorem 3.5 does it. I understand that complexity analysis for this algorithm may be non-trivial, but I believe it should be an essential part of the theoretical analysis. Therefore, I keep my score.
> > >
> > > Minor: for comparison, the proposed algorithm would be of much interest if it can scale well. So I suggest having more experiments with various problem sizes when comparing with Gurobi.

---

> > > > ### Author Response · Authors · 2023-11-21
> > > > **Heartiest thanks for the comments and suggestions.**
> > > >
> > > > Dear Reviewer 9DAk,
> > > >
> > > >  Thank you for your prompt response; your comments are highly valued and a source of inspiration for us.
> > > >
> > > >  $\textbf{Response for bound on running time:}$ Upon a thorough reevaluation of our computations, we have successfully quantified an expression for $T(\epsilon)$. Following a meticulous and detailed analysis, we have determined that for
> > > >  $$
> > > > 			M=M(A,b,R):= \sup_{u\in \bar B_R(0)} \| A^\top (A(u\circ u) - b) \|_2, \quad K= K(A,b,R) := \Big\lceil 8\max(1,M)\Big\rceil.
> > > > 			$$
> > > >    the running time $T(\epsilon)$ could be upper bounded by $$(\eta(\epsilon)^{-1}+1)(\lambda(\epsilon)^{-1}+1)K^{\gamma}$$ where $\gamma= C\|A(u_0\circ u_0)-b\|^2+1$ for some constant $C>0$. Here $\delta$ and $C$ are model dependent constant which does not necessarily grow with the size of the problem. In our response on the dependence of $\eta(\epsilon)$ and $\lambda(\epsilon)$, we have indicated that $\eta(\epsilon), \lambda(\epsilon) = O(\epsilon)$.
> > > >
> > > >    In other word, $T(\epsilon) = O(n^{\gamma}\epsilon^{-2})$ where $n$ is the model size and $\gamma$ is constant which depends on $A,b$ and initialization $u_0$. Recall that $u_0= exp(-c/2\lambda(\epsilon))$. Therefore, for $\epsilon\leq 1/(\max_i c_i\log n)$ and $\|A\|,\|b\|\leq 1$, $\gamma$ could be upper bounded as $$\gamma \leq C(\|A\|^2\|u_0\circ u_0\|^2+\|b\|^2)+1\leq C(1\cdot 1+1)+1 = 2C+1.$$
> > > >
> > > >  We would be very happy to add this in our main results of the revised manuscript. Furthermore, in the process of deriving $T(\epsilon)$, we got also bound on the linear rate $\rho$ which is another of your query. We are pleased that your inquiries have sparked a new set of developments on our end.
> > > >
> > > > $\textbf{Response on the minor comments about more simulation:}$ We completely agree that more simulations with different problem size would make our claim more stronger and we are preparing to add more simulation in the Appendix like as you have indicated.
> > > >
> > > > We would greatly appreciate it if you could consider increasing your score should our response meet the criteria of your inquiry.
> > > >
> > > > Thank You,
> > > >
> > > > Authors.

---

> > > > > ### Comment · Reviewer_9DAk · 2023-11-21
> > > > > **Thanks for the response**
> > > > >
> > > > > I am glad to see the complexity result and appreciate the authors' effort they put in the rebuttal phase. I have raised my score.

---

### Official Review · Reviewer_p2u6 · 2023-10-31

**Soundness:** 3 good
**Presentation:** 1 poor
**Contribution:** 2 fair
**Rating:** 5
**Confidence:** 4

**Summary:**

This manuscript studies the implicit bias of re-parametrized gradient descent in which the macroscopic learning rates are used. By leveraging the characterization of the implicit bias and the convex geometry of a linear program, they prove the linear convergence of GD on the linear regression problem under the quadratic parametrization. Importantly, they make minimal assumptions to prove their results.

**Strengths:**

The extension of the previous results [1,2] on reparametrized gradient descent/flow under minimal assumptions on data and macroscopic choice of learning rate is a nontrivial result. The analysis under this generality introduces additional complexity, but this work successfully establishes (linear) convergence for this setting.

[1] Woodworth, B.E., Gunasekar, S., Lee, J., Moroshko, E., Savarese, P.H., Golan, I., Soudry, D., & Srebro, N. (2019). Kernel and Rich Regimes in Overparametrized Models. ArXiv, abs/2002.09277.

[2] Even, M., Pesme, S., Gunasekar, S., & Flammarion, N. (2023). (S)GD over Diagonal Linear Networks: Implicit Regularisation, Large Stepsizes and Edge of Stability. ArXiv, abs/2302.08982.

**Weaknesses:**

The presentation can be further improved in several ways to enhance clarity and readability:
*  Firstly, I could not follow how the similarity between Algorithm 1 and the Sinkhorn algorithm is used in the paper.
* Additionally, to the best of my knowledge, the result in [1] proves a similar result in the manuscript in a fairly general setting too. I think it would be helpful for the readers if the authors could elaborate on these points more in their revised manuscript.

Another aspect that requires attention is the lack of characterization of the effect of using large step sizes in the paper. In particular, [1] studies the effect of large step size in the same setting on the generalization; however, this study does not provide insight about the consequence of using large step size.

[1] Even, M., Pesme, S., Gunasekar, S., & Flammarion, N. (2023). (S)GD over Diagonal Linear Networks: Implicit Regularisation, Large Stepsizes and Edge of Stability. ArXiv, abs/2302.08982.

**Questions:**

In the experiments, the authors show that for the large step size cases, re-parametrized GD converges faster than the exponentiated gradient algorithm (or mirror descent with the entropy potential). Can they comment on why this is the case and if their results imply anything in that direction?

---

> ### Author Response · Authors · 2023-11-18
> **Rebuttal (Thanks for the review and incisive questions. We truly appreciate your comments.)**
>
> $\textbf{Response on the comparison with the Sinkhorn algorithm:}$ Certainly, this is an intriguing question. We acknowledge that the elucidation of the comparison between our algorithm and the Sinkhorn algorithm might not be entirely clear. Our intention was to delineate the similarities and distinctions between these two algorithms by scrutinizing the update rules in each iteration. For instance,   we demonstrated in Section 2.3 that our algorithm behaves as row and column rescaling algorithms just like as the Sinkhorn algorithm.
>
> Similar to the Sinkhorn, our neural reparametrized gradient descent initialize the coupling matrix of OT at  $X^{(0)} := K$ where $K_{ij} := e^{- C_{ij}/\lambda }$. When the stepsize is small, like as in Sinkhorn, the gradient descent rescales the rows and columns of the coupling matrix. In other words, as the step size gets smaller, gradient descent iterates look like
>   $$X^{(k+1)} \approx D(1_n - g^{(k)}) X^{(k)} D(1_m - h^{(k)})  $$ where $g^{(k)}$ and $h^{(k)}$ are respectively the vectors of residuals of the row sums and column sums of the coupling matrix $X^{(k)}$.
> 	Hence the updates above can also be viewed as row and column rescalings, although with different rescaling rules than the Sinkhorn.  Given the parallels between the Sinkhorn algorithm and the updates in our gradient descent, it's reasonable to assert that our approach introduces a Sinkhorn-like algorithm for solving linear programming problems within the context of diagonal linear networks. As far as our knowledge extends, this novel application has not been documented in prior works.
>
> $\textbf{Resnpose on the comparison with other works on DLN:}$  Although our work bears certain similarities to the references [1] and [2] cited by the reviewer, the primary distinction lies in the diverse approaches to reparametrization. In [1] and [2], the authors have used the reparametrization $\beta= u\odot u - v\odot v$ in the basis pursuit problem while we use a slightly different reparametrization which leads to different final representations of the limit of the gradient descent in our case. However, in both cases, the limit of the gradient descent solve regularized $L_1$-norm minimization problem of $\beta$. But the nature of regularization is different in our work compared to [1] and [2]. Moreover, at the time of submitting our work, the rate of convergence result for gradient descent, as presented in [1] or [2], was not available. To the best of our knowledge, this result was initially introduced in our work. In the following, we discuss the effect of stepsize in our gradient descent in the light of [2] and [3].
>
> $\textbf{Response on the effect of stepsize:}$  As the reviewer rightly mentioned, the stepsize play a crucial role in shaping the quality of the algorithm. Theoretical thresholds for the step size necessary for global linear convergence are detailed in Theorem 3.5. However, recent investigations, exemplified in the reference [2] pointed out by the reviewer and the recently published paper in arxiv [3], scrutinize the impact of an increasing step size. These studies, especially the latter one, reveals that, as the step size amplifies, gradient descent in quadratic regression traverses distinct phases. Beyond the theoretical bounds, the loss function oscillates around a diminishing trend, signifying the emergence of the 'catapult' phase. Notably, during the catapult phase, the application of specific ergodic averaging methods has been demonstrated to enhance generalization error. Further escalation of the step size leads to the onset of the periodic phase, marked by periodic oscillations in the loss function without a discernible downward trend. Subsequent to the periodic phase, the system transitions into the chaotic phase, characterized by a lack of convergence signals, ultimately culminating in divergence. Given that our gradient descent iterates mimic those of the least square quadratic regression, we anticipate that the insights from [3] will be applicable to our framework. We would be pleased to incorporate a comprehensive discussion of this in the revised version.
>
> [3] Chen, X., Balasubramanian, K., Ghosal, P., Agrawalla, B. (2023). From Stability to Chaos: Analyzing Gradient Descent Dynamics in Quadratic Regression. ArXiV:  2310.01687
>
> $\textbf{Response on the faster rate compared to mirror gradient descent:}$ That is an excellent question. We posit that our algorithm outpaces mirror descent due to the tendency of mirror descent updates to overshoot frequently, resulting in unnecessary excursions near the limit for extended periods. Nevertheless, it remains plausible that a judicious selection of the step size in mirror descent, akin to our algorithm, could potentially enhance its speed.

---

> > ### Author Response · Authors · 2023-11-20
> > **update/feedback**
> >
> > Dear Reviewer p2u6,
> >
> > As the deadline for the discussion phase is fast approaching, we were curious if you had any feedback for our response. Please also let us know if you have any further questions. Thank you and looking forward to hearing from you.
> >
> > Sincerely,
> >
> > Authors.

---

> > > ### Comment · Reviewer_p2u6 · 2023-11-21
> > >
> > > Comparison with the Sinkhorn algorithm: As far as I can follow, to establish the similarity with the Sinkhorn algorithm, the authors use a well-known property the exponentiated gradient descent algorithm, i.e., $e^{\eta h} \approx (1+\eta h)$ for small $\eta$, see [Chapter 4.4, 1] and [Chapter 2.7, 2]. Therefore, I still cannot follow how the similarity pointed out in the manuscript provides us with new insight.
> > >
> > > For the other points, I thank the authors for their explanations. Overall, I still think that the manuscript requires more work to improve the presentation. Therefore, I keep my score the same.
> > >
> > > [1]. Kivinen, J., & Warmuth, M.K. (1997). Exponentiated Gradient Versus Gradient Descent for Linear Predictors. Inf. Comput., 132, 1-63.
> > >
> > > [2] Cesa-Bianchi, N., & Lugosi, G. (2006). Prediction, learning, and games.

---

### Official Review · Reviewer_xDbj · 2023-11-02

**Soundness:** 3 good
**Presentation:** 3 good
**Contribution:** 3 good
**Rating:** 6
**Confidence:** 2

**Summary:**

The authors prove linear convergence rates for the discrete and continuous versions of gradient on diagonal linear networks. In addition, they show that the continuous and discrete versions converge toward the solution of an entropically regularized linear problem.

**Strengths:**

The theoretical results are new and elegant, I am not aware of previous results connecting linear programming and diagonal linear networks.

**Weaknesses:**

- Section 2.2 and 2.3 are a little bit scientifically "loose", it draws some connections with other methods, but I am not sure there are proper theoretical results that can be extracted from these parts

- Experiments. I understand this is a theoretical paper, but I think the paper would have more impact with more extensive experiments. The current experimental section illustrates the linear convergence of the algorithm and has one comparison to the mirror descent. Maybe the authors could provide an experiment with optimal transport and compare the proposed method to the standard Sinkhorn algorithm

**Questions:**

- In the experiment section, the authors mention that the theoretical step size found is too conservative. This conservative step size problem can usually be overcome with coordinate descent-like methods, that use a larger coordinate-specific step size. In addition, the authors mention some previous results on coordinate descent for diagonal linear networks. I was wondering if it is possible to extend the theoretical results of the authors to coordinate descent.

---

> ### Author Response · Authors · 2023-11-18
> **Rebuttal (Thanks for the review and incisive questions. We truly appreciate your comments.)**
>
> $\textbf{Response on the use of co-ordinate descent:}$ We are very grateful to you evaluation. This is indeed a great question.
>      Certainly, the selection of the step-size is a subject that merits deeper exploration, and we intend to delve into this aspect, drawing insights from recent studies elucidating the impact of step size on gradient descent. Exploring unique step-sizes for coordinate-wise descent presents a fascinating avenue for investigation. We are optimistic about the potential acceleration of our algorithm through coordinate descent; nevertheless, the analysis of its convergence and rate appears to pose distinctive challenges compared to our current approach.
>
> $\textbf{Response on the experimental comparison with the Sinkhorn algorithm and other LP solver:}$ Thank you for the question! Actually we compared our method with standard Sinkhorn, but unfortunately, we found that Sinkhorn is better for OT -- it more directly uses the structure of the problem. Nonetheless, we found that our algorithm can be useful for general LPs where Sinkhorn method is not applicable. For instance, we have performed a new experiment, comparing our method with Gurobi and another ADMM-based method.  In particular, we simulate the data with the same data-generating process as in Section 4 of the paper, but with with $m = 500$ and $n = 50000$. Here is a brief comparison of the methods in this example: Gurobi, SCS and our algorithm achieve respectively 0, -0.971,0.005 primal gaps and 0, 2.21e-10, 4.17e-11 feasibility gaps in 16.77, 40.49 and 9.26 seconds.
>
> Here the primal gap is defined as $(c^\top \hat x - c^\top x^*) / c^\top x^*$, with $\hat x$ be the solution by an algorithm and $x^*$ is the optimal solution (by Gurobi). The feasibility gap is defined as $\| A \hat x - b \|_2^2 / \max\{1, \| b\|_2^2 \} $.
> First, it can be seen that our method is much faster than SCS,
> where our method computes a solution with a smaller primal gap in a much shorter time.
> Our method is also faster than Gurobi if a low-accuracy solution with a primal gap $0.005$ is satisfactory for the underlying application.
> Moreover, we note that Gurobi takes $10.11$ seconds for preprocessing, which is already similar to the runtime of our method.
>
> Admittedly, our method is not able to obtain high-accuracy solutions as fast as Gurobi, but it is useful for quickly obtaining an approximate solution. Moreover, since Algorithm 1 only involves matrix-vector multiplications, it is easily implemented in a GPU-acceleration setting, which can potentially outperform Gurobi for larger problems.

---

> > ### Author Response · Authors · 2023-11-20
> > **update/feedback**
> >
> > Dear Reviewer xDbj,
> >
> > As the deadline for the discussion phase is fast approaching, we were curious if you had any feedback for our response. Please also let us know if you have any further questions. Thank you and looking forward to hearing from you.
> >
> > Sincerely,
> >
> > Authors.

---

### Official Review · Reviewer_FLD4 · 2023-11-06

**Soundness:** 2 fair
**Presentation:** 3 good
**Contribution:** 2 fair
**Rating:** 6
**Confidence:** 3

**Summary:**

This work looks into the dynamics of gradient descent (or GD) optimization, specifically focusing on the reparameterized GD for linear programmings. By reparameterizing the problem, the author(s) claim to provide a clearer understanding of the implicit bias of GD, particularly how it induces sparsity in solutions. The paper's theoretical contributions demonstrate that this reparameterized GD converges to zero-loss solutions and biases the flow toward solutions with better sparsity properties than conventional vanilla GD. The author support their claims with mathematical proofs and experiments that compare the performance of reparameterized GD with traditional GD and mirror-descent methods, suggesting advantages in terms of sparsity and convergence.

**Strengths:**

This study's approach, reparameterizing GD for linear programs, offering a novel perspective on the implicit bias of GD. The quality is demonstrated through rigorous mathematical analyses, which include bounding the iterates of the algorithm and characterizing the limit points of the convergence.

Clarity is another strength, with the paper presenting its methodology and findings in a structured and understandable manner.

**Weaknesses:**

- a notable weakness is the limited scope of the experimental setup; their simulation relies on isotropic Gaussian features, which may not be representative of real datasets that often contain features with varying scales and correlations. Moreover, the paper does not discuss the impact of non-Gaussian noise or different initialization schemes, which could potentially affect the generalization of the results
  - can the author provide some results on real-world benchmarks? If conditions permit, I also suggest that the author compare it with sota linear programming algorithms (like some commercial solvers) and plot learning curves of the objective func value decreasing over time t

- the paper's analysis assumes a batch size of $m$, which may not scale well or apply directly to the common practice of using mini-batches. Moreover, the discussion on the impact of step size and batch size on the effective initialization scale is not sufficiently detailed, potentially limiting the applicability of their findings

- while the paper offers a comparison with mirror descent, it may benefit from a broader comparison with other optimization algorithms in the community (empirically, or theoretically) to establish a more comprehensive understanding of its advantages

**Questions:**

1. In practical applications, mini-batch GD is common; could the authors speculate on how their results might change with the introduction of mini-batches? Could the authors discuss the limitations of their assumptions regarding initialization and step sizes in more technical depth, possibly suggesting how these might be relaxed or generalized?

2. Are there any theoretical insights from the paper that could suggest practical guidelines for tuning hyperparameters (like the step-size) in GD to leverage the sparsity-inducing properties observed in the reparameterized model?

3. The theoretical framework is focused on diagonal linear networks; could the authors discuss the potential challenges and modifications required to extend their framework to deep / non-linear networks?

---

> ### Author Response · Authors · 2023-11-18
> **Rebuttal (Thanks for the review and incisive questions. We truly appreciate your comments.)**
>
> $\textbf{Response on the use of minibatch SGD:}$  We appreciate your inquiries. The substitution of minibatch gradient descent for gradient descent is not only feasible but can potentially offer advantages. Upon closer examination of our proofs and drawing on similarities between GD and one pass SGD on DLN, it becomes evident to us that the theoretical findings presented in our paper remain robust for minibatch GD with an appropriate choice of the batch size.
>
>  $\textbf{Response on the effect of initialization:}$ Exploring the impact of step size and initialization is indeed a crucial avenue of investigation. The initialization of our gradient descent proves to be a pivotal factor in guiding the algorithm toward the desired solution. As demonstrated in Theorem ~2.2 and 3.5, we established that initializing the gradient descent with
>       $u^0= \exp(-c/2\lambda)$ leads to convergence toward the solution of $$\min_u\{\langle c, u\odot u\rangle + \frac{\lambda}{1-\lambda}\sum_{i} u_i\odot u_i \log (u_i\odot u_i)\}.$$  Indeed, selecting a small positive value for
> $\lambda$ results in a very small initialization. This, in turn, enhances the quality of the solution, given that the regularization term diminishes as $\lambda$ approaches $0$.
>
> $\textbf{Response on the effects of stepsize:}$  The impact of the step size is pivotal in achieving a superior generalization error. The theoretical bounds on the step size required for global linear convergence have been established in Theorem. However, recent works, such as in [1], delve into the effect of an increasing step size. In [1], it is demonstrated that, as the step size increases, gradient descent in quadratic regression undergoes various phases. Beyond theoretical bounds, the loss function oscillates around a diminishing trend, indicating the onset of the 'catapult' phase. Notably, in the catapult phase, employing certain ergodic averaging methods has been shown to lead to improved generalization error. Further increasing the step size leads to entry into the periodic phase, characterized by periodic oscillations in the loss function with no discernible downward trend. Subsequently, the chaotic phase ensues, devoid of any signs of convergence, ultimately culminating in divergence. Given that our gradient descent iterates mirror those of least square quadratic regression, we anticipate that the observations made in [1] will hold for our framework. We would be delighted to incorporate a detailed discussion of this in the revised version.
>
> [1] Chen, X., Balasubramanian, K., Ghosal, P., Agrawalla, B. (2023). From Stability to Chaos: Analyzing Gradient Descent Dynamics in Quadratic Regression. ArXiV:  2310.01687
>
> $\textbf{Response on comparison with other commercial LP solver:}$  We have run a new experiment comparing our method against
>  the commercial LP solver Gurobi and another first-order method: an ADMM-based solver SCS. In particular, we simulate the data with the same data-generating process as in Section 4 of the paper, but with $m = 500$ and $n = 50000$. Here is a brief comparison of the methods in this example: Gurobi, SCS and our algorithm achieve respectively 0, -0.971,0.005 primal gaps and 0, 2.21e-10, 4.17e-11 feasibility gaps in 16.77, 40.49 and 9.26 seconds.
>
> Here the primal gap is defined as $(c^\top \hat x - c^\top x^*) / c^\top x^*$, with $\hat x$ be the solution by an algorithm and $x^*$ is the optimal solution (by Gurobi). The feasibility gap is defined as $\| A \hat x - b \|_2^2 / \max\{1, \| b\|_2^2 \} $.
> First, it can be seen that our method is much faster than SCS,
> where our method computes a solution with a smaller primal gap in a much shorter time.
> Our method is also faster than Gurobi if a low-accuracy solution with a primal gap $0.005$ is satisfactory for the underlying application.
> Moreover, we note that Gurobi takes $10.11$ seconds for preprocessing, which is already similar to the runtime of our method.
>
> Admittedly, our method is not able to obtain high-accuracy solutions as fast as Gurobi, but it is useful for quickly obtaining an approximate solution. Moreover, since Algorithm 1 only involves matrix-vector multiplications, it is easily implemented in a GPU-acceleration setting, which can potentially outperform Gurobi for larger problems.
>
> $\textbf{Response on generalization for deep/non-linear network:}$  Instead of a diagonal linear network, one might consider employing an $k$-layer deep diagonal linear network. It amounts to parametrizing the gradient descent as $x= u\odot \cdots \odot u$ over $k$ layers for $k>2$. The gradient descent initialized at $u_0 = \alpha1_n$ then $u_k\odot \cdots \odot u_k$ converges to the solution of $\min_x\{\langle 1_n, x\rangle - \frac{k \alpha^{k-2}}{2} \sum_{i=1}^d x_i^{2/k}\}$.  Extending this result to intricate architectures of deep non-linear networks is open. However, the optimal choice of initialization in such scenarios remains an area of active research.

---

> > ### Author Response · Authors · 2023-11-20
> > **update/feedback**
> >
> > Dear Reviewer FLD4,
> >
> > As the deadline for the discussion phase is fast approaching, we were curious if you had any feedback for our response. Please also let us know if you have any further questions. Thank you and looking forward to hearing from you.
> >
> > Sincerely,
> >
> > Authors.

---

> > > ### Comment · Reviewer_FLD4 · 2023-11-20
> > > **The rebuttal makes sense**
> > >
> > > I appreciate the authors' explanation, and will retain my positive score.

---

### Meta-Review · Area_Chair_tAMu · 2023-12-08

**Metareview:**

The authors propose to solve the problem of determining whether a linear program is feasible using a reparameterized gradient descent approach, and analyse a continuous time version of the approach showing that it relates to an entropy regularized solution to the linear program under technical assumptions. However, the limitations imposed by those assumptions is unclear, and further, the gap between the continuous time and practical discrete time formulations is unclear. Experimentally, the authors simply compare their approach against mirror descent, and ignore the vast literature on entropy regularized optimal transport algorithms. Given the limitations of the theoretical results and experiments, I recommend rejection.

**Justification For Why Not Higher Score:**

Neither theoretical nor empirical results justify the value of the approach proposed by the authors.

**Justification For Why Not Lower Score:**

N/A

---

### Decision · Program_Chairs · 2024-01-16

Reject